# On the Performance of Temporal Difference Learning With Neural Networks

**Haoxing Tian, Ioannis Ch. Paschalidis, Alex Olshevsky**
Department of Electrical and Computer Engineering
Boston University
Boston, MA 02215, USA
{tianhx, yannisp, alexols}@bu.edu

## Abstract

Neural Temporal Difference (TD) Learning is an approximate temporal difference method for policy evaluation that uses a neural network for function approximation. Analysis of Neural TD Learning has proven to be challenging. In this paper we provide a convergence analysis of Neural TD Learning with a projection onto $B(\theta_0, \omega)$, a ball of fixed radius $\omega$ around the initial point $\theta_0$. We show an approximation bound of $O(\epsilon + 1/\sqrt{m})$ where $\epsilon$ is the approximation quality of the best neural network in $B(\theta_0, \omega)$ and $m$ is the width of all hidden layers in the network.

## 1 Introduction

Temporal difference (TD) learning is considered to be a major milestone of reinforcement learning (RL). Proposed by Sutton (1988), TD Learning uses the Bellman error, which is a difference between an agent's predictions in a Markov Decision Process (MDP) and what it actually observes, to drive the process of learning an estimate of the value of every state.

To deal with large state-spaces, TD learning with linear function approximation was introduced in Tesauro (1995). A mathematical analysis was given in Tsitsiklis & Van Roy (1996), which shows the process converges under appropriate assumptions on step-size and sampling procedure. However, with nonlinear function approximation, TD Learning is not guaranteed to converge, as observed in Tsitsiklis & Van Roy (1996) (see also Achiam et al. (2019) for a more recent treatment).

Nevertheless, TD with neural network approximation, referred to as Neural TD, is used in practice despite the lack of strong theoretical guarantees. To our knowledge, rigorous analysis was only addressed in the three papers Cai et al. (2019); Xu & Gu (2020); Cayci et al. (2021).

In Cai et al. (2019), a single hidden layer neural architecture was considered along with projection on a ball around the initial condition; approximate convergence was proved to be an approximate stationary point of a certain function related to the linearization around the initial point. This result was generalized to multiple hidden layers in Xu & Gu (2020), but this generalization required projection on a ball of radius of $\omega \sim m^{-1/2}$ around the initial point, where $m$ is the width of the hidden layers. Because the radius of this projection goes to zero with $m$, this effectively fixes the neural network to a small distance from its initial condition. Both Cai et al. (2019); Xu & Gu (2020) additionally required certain regularity conditions on the policy. Finally, Cayci et al. (2021) gave a convergence result for a single hidden layer, also with a projection onto a radius of $\omega \sim m^{-1/2}$ around the initial point, but with the final objective being the representation error of neural approximation without any kind of linearization. This result also required a condition on the representability of the value function of the policy in terms of features from random initialization.

In this paper, we analyze Neural TD with a projection onto $B(\theta_0, \omega)$, a ball of fixed radius $\omega$ around the initial point $\theta_0$. We show an approximation bound of $O(\epsilon + 1/\sqrt{m})$ where $\epsilon$ is the approximation quality of the best neural network in $B(\theta_0, \omega)$. Our result improves on previous work because it does not require taking the radius $\omega$ to decay with $m$, does not make any regularity or representability assumptions on the policy, applies to any number of hidden layers, and bounds the error associated with the neural approximation without any kind of linearization around the initial condition.

The main technical difference between our paper and previous works is the choice of norm for analysis. We will describe this at a more technical level in the main body of the paper, but we use a norm introduced by Ollivier (2018), which is a convex combination of the usual $l_2$-norm weighted by the stationary distribution of the policy with the so-called Dirichlet semi-norm. The later has previously been used in the convergence analysis of Markov chains (Diaconis & Saloff-Coste (1996); Levin & Peres (2017)). It was shown in Ollivier (2018) that Neural TD is exactly gradient descent on this convex combination of norms if the underlying policy is reversible.

In the case where the policy is not reversible, these results were partially generalized in Liu & Olshevsky (2021), where it was shown that TD Learning with linear function approximation can be viewed as a so-called *gradient splitting*, a process which is analogous to gradient descent. We build heavily on that interpretation here. Our main technical argument is that Neural TD approximates the gradient splitting process at each step so that despite the nonlinearity of the approximation, an improvement in approximation quality can be guaranteed unless the system is already close to the best possible approximation over the projection radius. Notably, our arguments do not imply that the neural network stays close to its initialization and our empirical simulations show a significant benefit from taking the projection radius $\omega$ *not* to decay with the width $m$.

## 2 PRELIMINARIES

### 2.1 MARKOV DECISION PROCESSES

In this section, we present key concepts from MDPs, mostly to introduce our notation.

A finite discounted-reward MDP can be described by a tuple $(S, A, P_{\text{env}}, r, \gamma)$, where $S = \{s_1, s_2, \ldots, s_n\}$ is a finite state-space whose elements are vectors; $A$ is a finite action space; $P_{\text{env}} = (P_{\text{env}}(s'|s, a))_{s,s' \in S, a \in A}$ is the transition probability matrix, where $P_{\text{env}}(s'|s, a)$ is the probability of transitioning from $s$ to $s'$ after taking action $a$; $r : S \times A \to R$ is the reward function; and $\gamma \in (0, 1)$ is the discount factor. A policy $\pi$ in an MDP is a mapping $\pi : S \times A \to [0, 1]$ such that $\sum_{a \in A} \pi(s, a) = 1$ for all $s \in S$, where $\pi(s, a)$ is the probability that the agent takes action $a$ in state $s$. We will use $n$ for the number of states, i.e., $|S| = n$.

Given a policy $\pi$, we define the corresponding transition probability matrix $P = (P(s'|s))_{s,s' \in S}$ as

$$P(s'|s) = \sum_{a \in A} \pi(s, a) P_{\text{env}}(s'|s, a).$$

We also define the state reward function as

$$r(s) = \sum_{a} \pi(s, a) r(s, a).$$

Although $P$ and $r(s)$ depend on the policy $\pi$, throughout this paper the policy will be fixed and hence we will suppress this dependence in the notation.

The stationary distribution $\mu$ corresponding to the policy $\pi$ is defined to be a nonnegative vector with coordinates summing to one and satisfying $\mu^T = \mu^T P$. The Perron-Frobenius theorem guarantees that such a $\mu$ exists and is unique subject to some conditions on $P$, e.g., aperiodicity and irreducibility (Gantmacher (1964)). We use $\mu(s)$ to denote each entry of $\mu$.

The value function of the policy $\pi$ is defined as:

$$V^*(s) = \mathbb{E}_{s,a \sim \pi} \left[ \sum_{t=0}^{+\infty} \gamma^t r(s^t) \right],$$

where $\mathbb{E}_{s,a \sim \pi}$ stands for the expectation when the starting state is $s$ and actions are taken according to policy $\pi$, and $s^t$ is the $t$'th state encountered. Note that this quantity depends on $\pi$, but once again we suppress this dependence because the policy will be fixed throughout this paper. Moreover, note that, despite the star superscript, $V^*$ is not the optimal value function but rather the true value function corresponding to policy $\pi$.

It is well known that this value function satisfies the Bellman equation,

$$V^* = R + \gamma P V^*, \tag{1}$$

where $V^*$ is a vector whose $i$'th element is $V^*(s_i)$ and $R$ is a vector whose $i$'th element is $r(s_i)$.

We will further assume that rewards are bounded in $[-r_{\max}, r_{\max}]$ as in the following assumption.

**Assumption 2.1.** *For any $s, a \in S \times A$, we have $|r(s,a)| \leq r_{\max}$.*

This immediately implies that

$$|V^*(s)| \leq \frac{r_{\max}}{1 - \gamma}, \ \forall s \in S. \tag{2}$$

## 2.2 Markov Chain Noise Model

There are two standard sampling models where policy evaluation methods are usually considered. The simplest model involves i.i.d. sampling of $s_t$ at each step from stationary distribution $\mu$.

Alternatively, in the Markov model the states $s_t$ are collected from a single path of Markov chain transitioning according to $P$. It is still assumed in this case that the initial distribution is $\mu$, so that the distribution of each $s_t$ is still $\mu$, with the difference that the successive states are now not independent. This can always be approximately satisfied by generating a sufficiently long path from $P$ and ignoring the initial states.

Assuming that the underlying Markov chain mixes with a geometric rate is common in many analyses, like Bhandari et al. (2018); Liu & Olshevsky (2021). We will also make this assumption. Formally, let $P^t$ denote the matrix $P$ raised to the $t$'th power, $P_{s,:}^t$ be the row of $P^t$ corresponding to state $s$, and $||\cdot||_{\text{TV}}$ the total variation distance.

**Assumption 2.2.** *There exists constant $C > 0$ and $\beta \in (0,1)$ such that*

$$\max_s ||P_{s,:}^t - \mu||_{\text{TV}} \leq C\beta^t.$$

This assumption guarantees "mixing": no matter the initial distribution, the state will get closer and closer to the stationary distribution $\mu$ as $t$ increases. As pointed out in Levin & Peres (2017), this assumption always holds when the Markov chain is irreducible and aperiodic. Another useful quantity is the mixing time, $\tau_{\text{mix}}(\epsilon_{\text{mix}})$, defined as the smallest integer $t$ such that

$$\max_s ||P_{s,:}^t - \mu||_{\text{TV}} \leq \epsilon_{\text{mix}}.$$

Note that Assumption 2.2 implies

$$\tau_{\text{mix}}(\epsilon_{\text{mix}}) = \log_\beta \frac{\epsilon_{\text{mix}}}{C}. \tag{3}$$

For simplicity, we will use $\tau_{\text{mix}}$ without specifying its dependence on $\epsilon_{\text{mix}}$ throughout the paper.

## 2.3 $D$-norm and Dirichlet Norm in MDPs

We now introduce the so-called $D$-norm $||\cdot||_D$ and the Dirichlet semi-norm $||\cdot||_{\text{Dir}}$ associated with a policy. While the former has long been used for the analysis of temporal difference learning dating back to Tsitsiklis & Van Roy (1996), the latter has, to our knowledge, been introduced in the context of RL relatively recently in Ollivier (2018).

Let $D = \text{diag}(\mu(s))$ be the diagonal matrix whose elements are given by the entries of the stationary distribution $\mu$. Given a function $f : S \to R$, its $D$-norm is defined as

$$||f||_D^2 = f^T D f = \sum_{s \in S} \mu(s)f(s)^2. \tag{4}$$

The $D$-norm is similar to the Euclidean norm except each entry is weighted proportionally to the stationary distribution. We also define the Dirichlet semi-norm of $f$:

$$||f||_{\text{Dir}}^2 = \frac{1}{2} \sum_{s,s' \in S} \mu(s)P(s'|s)(f(s') - f(s))^2. \tag{5}$$

A semi-norm satisfies the triangle inequality and absolute homogeneity, as any norm, but it is not a norm as it may be equal to zero at a non-zero vector. Note that $||f||_{\text{Dir}}$ depends on the policy both through the stationary distribution $\mu(s)$ as well as through the transition matrix $P$.

Finally, following Ollivier (2018), the weighted combination of the $D$-norm and the Dirichlet semi-norm is denoted as $\mathcal{N}(f)$ and defined as

$$\mathcal{N}(f) = (1-\gamma)||f||_D^2 + \gamma||f||_{\text{Dir}}^2.$$

Note that $\sqrt{\mathcal{N}(f)}$ is a valid norm: since $\mathcal{N}(f)$ is quadratic, we can write $\mathcal{N}(f) = f^T N f$ for some symmetric matrix $N$; examining the first term in the definition of $\mathcal{N}(f)$ we see that $N \succeq (1-\gamma)\text{diag}(\pi_1, \ldots, \pi_n) \succ 0$ by irreducibility and aperiodicity.

## 2.4 NEURAL NETWORK BASED APPROXIMATION

In this section we closely follow the notation from the previous works in Cai et al. (2019); Allen-Zhu et al. (2019); Liu et al. (2020); Ollivier (2018) on neural approximations. We define a multi-layer fully connected neural network by the following recursion:

$$x^{(k)} = \frac{1}{\sqrt{m}}\sigma\left(\theta^{(k)}x^{(k-1)}\right), \text{ for } k \in \{1, \ldots, K\},$$

where $\sigma$ is an activation function and the input is state of the MDP: $x^{(0)} \in S$. Next, we define

$$V(s, \theta) = \frac{1}{\sqrt{m}}b^T x^{(K)},$$

to be the output with no activation function on the output. We assume that each entry of $\theta^{(k)}$ are initialized from $N(0, 1)$ and each entry of the vector $b$ satisfies $|b_r| \leq 1, \forall r$. We further assume that all the hidden layers have the same width which we denote by $m$, i.e., all the matrices $\theta^{(k)}$ have first dimension of $m$. Note that the total number of layers in the neural network is denoted by $K$.

We will stack up the weights of different layers into a column vector $\theta$ consisting of the entries of the matrices $\theta^{(1)}, \ldots, \theta^{(K)}$, with norm defined by

$$||\theta||^2 = \sum_{k=1}^{K} ||\theta^{(k)}||_F^2,$$

where $||\cdot||_F$ is the Frobenius norm. During the training process, only the weights $\theta$ will be updated while the weights $b$ will be left to their initial value.

This particular definition of a neural network, as well as the decision to leave $b$ fixed, is used by many papers on both TD Learning (i.e., Xu & Gu (2020)) and Deep Neural Network (i.e., Liu et al. (2020)) analysis. Although there is not an explicit bias term above, this definition does allow each layer to have different bias. This can be reached by setting the last entry of $x^{(0)}$ to be 1 and last row of each $\theta^{(k)}$ to be $(0, \ldots, 0, 1)$.

We can view this neural network as mapping the parameters $\theta$ to a vector with as many entries as the number of states. Specifically, we can define the vector $V(\theta)$ whose $i$'th entry is

$$[V(\theta)]_i = V(s_i, \theta), \forall i \in 1, 2, \ldots, n.$$

Note that $n$ in the above equation denotes the number of states in the MDP, i.e., $|S|$. We remark that this vector will never be actually used in the execution of any algorithms we discuss due to its large size, but its use is still useful conceptually. The Jacobian of $V(\theta)$ is then the matrix

$$\nabla_\theta V(\theta) = \begin{bmatrix} \nabla_\theta V(s_1, \theta) \\ \vdots \\ \nabla_\theta V(s_n, \theta) \end{bmatrix},$$

where, abusing standard notation, $\nabla_\theta V(s, \theta)$ are defined to be row vectors.

A standard assumption made to simplify the analysis is the following.

**Assumption 2.3.** *Suppose for any $i \in \{1, 2, \ldots, n\}$, $||s_i|| \leq 1$ where $||\cdot||$ stands for the $l_2$-norm.*

Assumption 2.3 can always be satisfied by scaling, as we typically have control of how we choose to represent the states of the MDP. It is also a common assumption that appears in many previous works (e.g., Cai et al. (2019); Allen-Zhu et al. (2019)).

A function $f$ is called $L$-Lipschitz if
$$|f(x) - f(y)| \le L|x - y|, \ \forall x, y.$$
And a differentiable function $f : \mathbb{R} \to \mathbb{R}$ is $c_0$-smooth if
$$|f'(x) - f'(y)| \le c_0|x - y|, \ \forall x, y,$$
where $f'$ stands for the derivative of $f$. We find it helpful to make the following assumption:

**Assumption 2.4.** *The activation function $\sigma$ is $l$-Lipschitz and $c_0$-smooth.*

The smoothness condition implies that our results below are not directly applicable to popular functions like ReLU. However, many activation functions are twice differentiable (e.g., sigmoid, tanh, arctan, softplus) and one could always use a smooth approximation to a ReLU activation (e.g., GeLU or ELU). We do need the smoothness of the input-to-output map as we will use the result from Liu et al. (2020), which claims the neural network is $O(\frac{1}{\sqrt{m}})$-smoothness with respect to its parameters. The following assumption is also required to implement their result:

**Assumption 2.5.** *For any $k \in \{1, 2, \ldots, K\}$, given $i \in \{1, 2, \ldots, m\}$, there exists some constant $c^{(k)} > 0$, such that $|x_i^{(k)}| = \tilde{O}(1)$ at initialization. Here, $x_i^{(k)}$ means the $i$'th entry of $x^{(k)}$.*

### 2.5 NEURAL TD

In this section, we introduce (projected) neural TD learning. At each time step $t$, this algorithm samples state $s$ from the stationary distribution from either of two sampling models discussed earlier, generates the next state $s'$ in the MDP, and computes the temporal difference error, defined as
$$\delta_t = r(s) + \gamma V(s', \theta_t) - V(s, \theta_t).$$
Defining $g(\theta_t)$ as
$$g(\theta_t) = \nabla_\theta V(s, \theta_t)\delta_t,$$
projected neural TD updates the weights as
$$\begin{aligned} \theta_{t+1/2} &= \theta_t + \alpha_t g(\theta_t), \\ \theta_{t+1} &= \mathbf{Proj}(\theta_{t+1/2}), \end{aligned} \tag{6}$$
where the projection is onto a ball of radius $\omega$ around the initial condition:
$$\mathbf{Proj}(\theta) = \arg\min_{x:||x-\theta_0||\le\omega} ||x - \theta||.$$

Projection is a common tool in neural TD and neural Q-learning to try to stabilize the iterates, since divergence can occur (Achiam et al. (2019); Van Hasselt et al. (2018)). Most analyses of TD learning proceed by comparing the evolution of TD to the mean-path update, defined as
$$\begin{aligned} \theta_{t+1/2} &= \theta_t + \alpha_t \bar{g}(\theta_t), \\ \theta_{t+1} &= \mathbf{Proj}(\theta_{t+1/2}), \end{aligned}$$
where
$$\begin{aligned} \bar{g}(\theta_t) &= \mathbb{E}[g(\theta_t)|\theta_t] \\ &= \sum_s \mu(s)\nabla_\theta V(s, \theta_t)\mathbb{E}_{s'|s}[r(s) + \gamma V(s', \theta_t) - V(s, \theta_t)] \\ &= \nabla_\theta V(\theta_t)^T D(R + \gamma PV(\theta_t) - V(\theta_t)). \end{aligned}$$
It is convenient to rewrite $\bar{g}(\theta_t)$ in terms of the difference between $V(\theta_t)$ and $V^*$, which can be done by subtracting Eq.(1) with the result being
$$\bar{g}(\theta_t) = \nabla_\theta V(\theta_t)^T D(\gamma P - I)(V(\theta_t) - V^*). \tag{7}$$
For simplicity, for the rest of paper we will define $\Theta$ to be the set onto which we are projecting:
$$\Theta = B(\theta_0, \omega) = \{\theta \mid ||\theta - \theta_0|| \le \omega\}.$$
Finally, we will say that $\hat{V}^*$ *is an $\epsilon$-approximation to the true value function $V^*$ if*
$$\max_{s\in S} |\hat{V}^*(s) - V^*(s)| \le \epsilon. \tag{8}$$
We will denote $\hat{\theta}_*$ to be the point where the function $V(\hat{\theta}_*) = \hat{V}^*$ is the $\epsilon$-approximation of $V^*$.

## 3   OUR MAIN RESULT

We can now state the main contribution of this paper, which is a performance result for Neural TD. We will require an assumption to the effect that the initialization is not too large.

**Assumption 3.1.** *For all $k \in \{1, 2, \ldots, K\}$, $\|\theta_0^{(k)}\| \leq O(\sqrt{m})$.*

This is a common assumption: theoretical justification can be found in Liu et al. (2020). In particular, this assumption holds with high probability for a random Gaussian initialization if, for example, $K$ (the depth) grows slower than exponentially in $m$ (the width).

Now we are ready to state our result.

**Theorem 3.1.** *Suppose Assumption 2.1, 2.2, 2.3, 2.4, 2.5, 3.1 hold, $\theta_t$ is generated by projected Neural TD with the constant step-size $\alpha_t = \alpha$, and $C, \beta$ come from the Assumption 2.2.*

*(a) Under i.i.d. sampling, we have*

$$\frac{1}{T}\sum_{t=0}^{T-1} \mathbb{E}[\mathcal{N}(V(\theta_t) - V(\hat{\theta}_*))] \leq \frac{\|\theta_0 - \hat{\theta}_*\|^2}{2\alpha T} + O\left(\epsilon + \frac{1}{\sqrt{m}}\right) + O\left(\alpha\epsilon^2\right) + \frac{1}{(1-\gamma)^2}O\left(\alpha\right).$$

*In particular, if $\alpha = T^{-1/2}$ and $\epsilon \leq 1$, we have*

$$\frac{1}{T}\sum_{t=0}^{T-1} \mathbb{E}[\mathcal{N}(V(\theta_t) - V(\hat{\theta}_*))] \leq \frac{\|\theta_0 - \hat{\theta}_*\|^2}{2\sqrt{T}} + O\left(\epsilon + \frac{1}{\sqrt{m}}\right) + \frac{1}{(1-\gamma)^2}O\left(\frac{1}{\sqrt{T}}\right).$$

*(b) Under Markov sampling, we have*

$$\frac{1}{T}\sum_{t=0}^{T-1} \mathbb{E}[\mathcal{N}(V(\theta_t) - V(\hat{\theta}_*))] \leq \frac{\|\theta_0 - \hat{\theta}_*\|^2}{2\alpha T} + O\left(\epsilon + \frac{1}{\sqrt{m}}\right) + O\left(\alpha\epsilon^2\right) + \frac{1}{(1-\gamma)^2}O\left(\alpha\right)$$

$$+ O\left(\alpha\frac{\log\frac{C}{\alpha}}{1-\beta}\epsilon^2\right) + \frac{1}{(1-\gamma)^2}O\left(\alpha\frac{\log\frac{C}{\alpha}}{1-\beta}\right),$$

*In particular, if $\alpha = T^{-1/2}$ and $\epsilon \leq 1$, we have*

$$\frac{1}{T}\sum_{t=0}^{T-1} \mathbb{E}[\mathcal{N}(V(\theta_t) - V(\hat{\theta}_*))] \leq \frac{\|\theta_0 - \hat{\theta}_*\|^2}{2\sqrt{T}} + O\left(\epsilon + \frac{1}{\sqrt{m}}\right) + \frac{1}{(1-\gamma)^2}O\left(\frac{1}{\sqrt{T}}\right)$$

$$+ \frac{1}{(1-\gamma)^2}O\left(\frac{1}{1-\beta}\frac{\log(C\sqrt{T})}{\sqrt{T}}\right).$$

In all $O(\cdot)$ notations above, we treat factors that do not depend on $T, \epsilon, m, \alpha, \beta, \theta_0$ as constants.

As mentioned earlier, the key distinguishing feature of this theorem is the choice of norm. The left-hand side of all the equations measures the difference between $V(\theta_t)$ and the best possible $V(\hat{\theta}_*)$ within $B(\theta_0, \omega)$ by taking the $\mathcal{N}(\cdot)$ norm.

We note that since, trivially, $\|f\|_D^2 \leq \mathcal{N}(f)/(1 - \gamma)$, where $\gamma$ is the discount factor, one can simply replace the left-hand sides of all the equations by $\|f\|_D^2$ to obtain results that look more similar to the previous literature on TD, which usually proceeds based on an analysis in the $D$-norm (e.g., Tsitsiklis & Van Roy (1996)).

As is common in analyses of SGD, the performance measure is the average of performance measures from 1 to $T$. If a particular $\theta_{t'}$ is sought that satisfies the bounds obtained, a standard trick is to choose $t'$ to be uniform from $1, \ldots, T$. In that case, the expectation $E[\mathcal{N}(V(\theta_{t'}) - V(\hat{\theta}_*)]$ is exactly the left-hand side of all the bounds in the theorem.

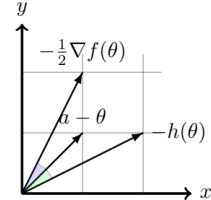

Figure 1: Key property of gradient splitting: $h(\theta)$ has the same inner product with $a - \theta$ as $(1/2)\nabla f(\theta)$.

Looking at the right-hand sides of all the equations, the theorem guarantees a final error of $O(\epsilon + 1/\sqrt{m})$, with a convergence rate that scales as $\tilde{O}(1/\sqrt{T})$. The difference between the two cases is that that the Markov sampling case contains an additional term containing the mixing time. As a result of this extra term, the convergence time in the Markov case is worse by a factor of $O(\log \sqrt{T})$.

We now revisit the discussion of the novelty of this paper. First, in contrast to previous work, we do not assume the projection radius $\omega$ has to decay with $m$, nor do we restrict our analysis to a single hidden layer case. In our proof, the projection radius appears as a constant in the $O(\cdot)$ notation, which is why we need to assume it is a constant. Second, the left-hand side of the equations is a measure of the error $V(\theta_t) - V(\hat{\theta}_*)$, which can be thought of the difference between the error of the neural network and the

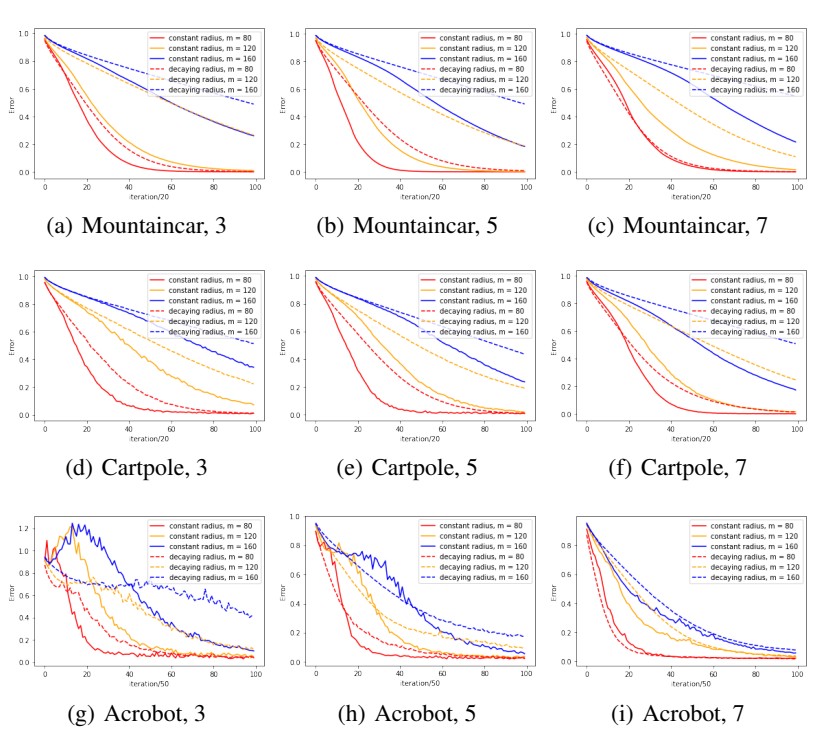

(a) Mountaincar, 3    (b) Mountaincar, 5    (c) Mountaincar, 7

(d) Cartpole, 3    (e) Cartpole, 5    (f) Cartpole, 7

(g) Acrobot, 3    (h) Acrobot, 5    (i) Acrobot, 7

Figure 2: Averaged Bellman error.

best possible error in $B(\theta_0, \omega)$. Moreover, as already discussed, the left-hand side of the equations above is greater than the quantity

$$(1-\gamma)\|V(\theta_t) - V(\hat{\theta}_*)\|^2_D = (1-\gamma)\sum_s \mu(s)(V(s,\theta_t) - V(s,\hat{\theta}_*))^2,$$

which is a natural measure of the average error. The point is that the network is not being linearized around the initial condition in any sense. Finally, no additional assumptions on the policy are being made here; in particular, no assumptions that the policy is regular or that it is representable are necessary. As outlined in the Introduction, out of the four potentially undesirable elements discussed here (small projection radius, linearization around the initial condition, assumptions on policy, restriction to single layer case) all previous papers suffered from at least three.

## 4 DISTINGUISHING FEATURE OF OUR ANALYSIS

The main technical difference between our work and the previous papers is the use of the function $\mathcal{N}(\cdot)$ to measure the approximation error. Here, we follow Ollivier (2018); Liu & Olshevsky (2021) which explained why this function is the "right" function to analyze policy evaluation.

In Ollivier (2018) it was shown that if the matrix $P$ corresponds to a reversible Markov chain, then $E[\bar{g}(\theta_t)] = \nabla_\theta \mathcal{N}(f)$ for some $f$. This makes neural TD very easy to analyze, as it can be viewed as gradient descent. Unfortunately, in practice policies are almost never reversible.

In Liu & Olshevsky (2021), it was shown how to further use the function $\mathcal{N}(\cdot)$ to analyze TD with linear approximation when the policy is not necessarily reversible. The key idea was the notion of a

gradient splitting: a linear function $h(\theta)$ is said to be a gradient splitting of a convex quadratic $f(\theta)$ minimized at $\theta = a$ if

$$\frac{1}{2}\nabla f(\theta)^T(a-\theta) = h(\theta)^T(\theta-a). \tag{9}$$

In other words, $h(\theta)$ has exactly the same inner product with the "direction to the optimal solution" as the true gradient of $f(\theta)$ (up to the factor of $1/2$). The significance of this is that it allows many analyses of gradient descent to be modified to analyze TD with linear approximation, since the key step in analyses of gradient descent is usually to argue that the left-hand side of Eq. (9) is negative, signifying that gradient descent "makes progress" towards the optimal solution. In Liu & Olshevsky (2021) it was shown that for TD with linear approximation, $\bar{g}(\theta)$

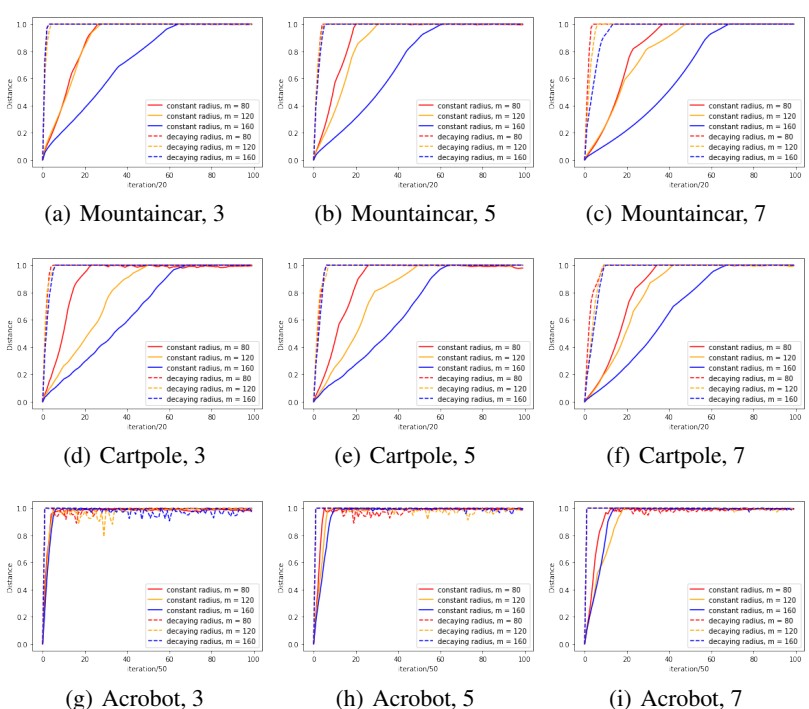

(a) Mountaincar, 3     (b) Mountaincar, 5     (c) Mountaincar, 7

(d) Cartpole, 3     (e) Cartpole, 5     (f) Cartpole, 7

(g) Acrobot, 3     (h) Acrobot, 5     (i) Acrobot, 7

Figure 3: Distance to initialization divided by the projection radius.

is exactly the gradient splitting of $\mathcal{N}(f)$. We build on that idea in this paper as follows. We use recent NTK-style bounds from Liu et al. (2020) to argue that with increasing width, the neural approximation gets more linear. For finite $m$, we can then modify existing techniques for analyzing gradient descent with errors in the gradient evaluations to analyze the neural TD update, which we view as *gradient splitting with error in the evaluations.*

It should be stressed that the most interesting "regime" to which we expect these results to be applicable is when $m$ is only moderately large. Intuitively, when $m$ is too large, the network will be very close to linear, and the benefit over taking random linear features will be small. In terms of our bounds, in this case we would expect the $\epsilon$ term in Theorem 3.1 to be large. On the other hand, when $m$ is too small, the error bound scaling with width of $m^{-1/2}$ will not be very attractive. On the other hand, our results can be applied to the "middle range" – say $m \sim 100$ – where, depending on the context of the problem, an $\sim m^{-1/2}$ error is acceptable. Informally, in this regime the network will be sufficiently linear to converge well, but not so linear to lose approximation quality. Our simulations confirm this, as we verify good approximations for networks of approximately this width on several open AI benchmarks.

## 5 SIMULATIONS

It might be objected that many analysis of neural network training in the large-width regime proceed by arguing that the neural network stays around its initial point (e.g., Chizat et al. (2019); Telgarsky (2020)). If so, then the part of our results dealing with avoiding a projection radius that goes to zero with $m$ would not make much of a difference in practice. Here, we show empirically that, setting the projection radius $\omega$ to be a constant rather than $\omega \sim m^{-1/2}$ does make a substantial difference.

We produce simulations on Open AI Gym tasks: Mountaincar, Cartpole and Acrobot. In each task, a trained policy is used to sample data to train a fully connected neural network for policy evaluation using Projected Neural TD learning. The policy in Mountaincar and Cartpole is trained by Proximal Policy Optimization (PPO) (Schulman et al. (2017)) while in Acrobot it is trained by Deep Q-Learning (DQN) (Mnih et al. (2015)). Both algorithms are implemented through the Stable Baselines package Raffin et al. (2021).

We run a total of 2000 (5000 in Acrobot task) steps on each task, computing the $y$ value during the training process every 20 (50 in Acrobot task) step. Different widths $m$ are chosen for each task. In each figure, the $x$-axis plots time steps while the $y$-axis plots the averaged Bellman error, the distance to initialization (given by $\frac{||\theta_t-\theta_0||}{\omega}$), and the difference between gradients (given by $||\nabla_\theta V(s_t,\theta_t) - \nabla_\theta V(s_t,\theta_0)||$), respectively. Each subtitle of each figure suggests which task is performed on and how many hidden layers are used. It is clear from the figures that the networks we use are nonlinear. It is also clear that networks with a decaying projection radius are significantly outperformed by networks with a constant projection radius.

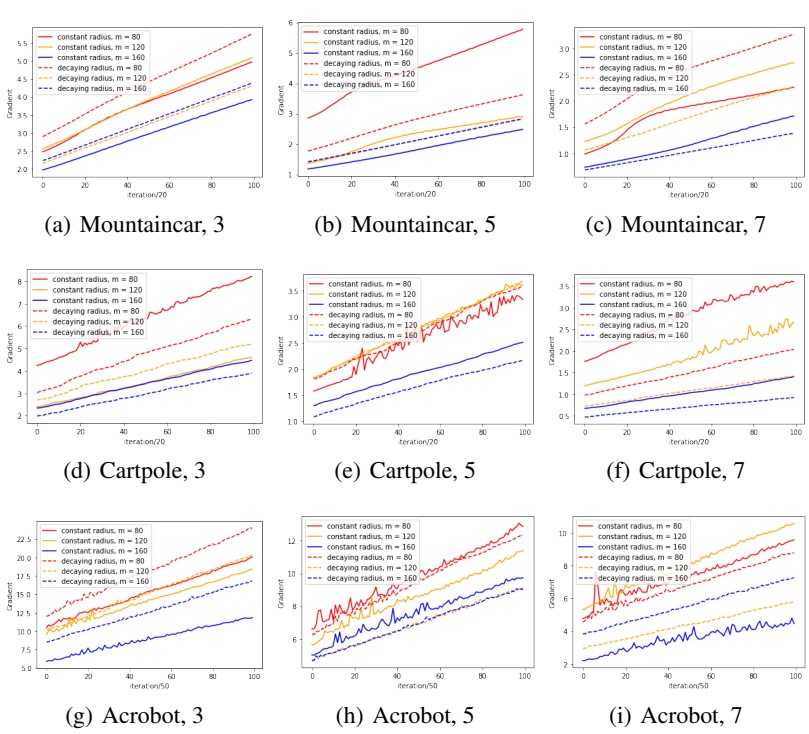

Figure 4: Gradient differences $||\nabla_\theta V(s_t,\theta_t) - \nabla_\theta V(s_t,\theta_0)||$ as a measure of nonlinearity of the neural network.

## 6 CONCLUSIONS

We have provided an analysis of Projected Neural TD for policy evaluation. We have shown that if the projection set is a ball of constant radius $B(\theta_0,\omega)$ around the initial point $\theta_0$, the final approximation error is $O(\epsilon + 1/\sqrt{m})$, where $\epsilon$ is the approximation quality of the best neural network in $B(\theta_0,\omega)$ and $m$ is the width of all hidden layers in the network. Our result improves on previous works because it does not require taking the radius $\omega$ to decay with $m$, does not make any regularity or representability assumptions on the policy, applies to any number of hidden layers, and bounds the error associated with the neural approximation without any kind of linearization around the initial condition. We conjecture that unprojected neural TD converges to the optimal solution with high probability provided it begins within a radius of $O(\sqrt{m})$ away from a point which exactly describes the true value function. This conjecture is proved for single hidden layer networks. Finally, we have demonstrated empirically that these changes can make a substantial difference in performance.

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

# A   PROOF FOR THE MAIN RESULT

## A.1   USEFUL LEMMAS

Before we go into details, we first introduce the following lemmas.

Recall we have defined $\mathcal{N}(f)$ as

$$\mathcal{N}(f) = (1 - \gamma)||f||_D^2 + \gamma||f||_{\text{Dir}}^2.$$

The first lemma implies an important property of $\mathcal{N}(\cdot)$.

**Lemma A.1.** *For any function $f$ defined on the state space $S$, the following equation holds:*

$$-\mathcal{N}(f) = f^T D(\gamma P - I)f.$$

*Proof.* We can perform the following sequence of manipulations:

$$
\begin{aligned}
||f||_{\text{Dir}}^2 &= \frac{1}{2} \sum_{s,s'} \mu(s)P(s'|s)[f(s) - f(s')]^2 \\
&= \frac{1}{2} \sum_s \mu(s)f(s)^2 + \frac{1}{2} \sum_{s,s'} \mu(s)P(s'|s)f(s')^2 - \sum_{s,s'} \mu(s)P(s'|s)f(s)f(s') \\
&= \frac{1}{2} \sum_s \mu(s)f(s)^2 + \frac{1}{2} \sum_{s'} \mu(s')f(s')^2 - \sum_{s,s'} \mu(s)P(s'|s)f(s)f(s') \\
&= ||f||_D^2 - \sum_{s,s'} \mu(s)P(s'|s)f(s)f(s'),
\end{aligned}
$$

where the first equation uses Eq.(5) and the forth equation uses Eq.(4). Thus,

$$
\begin{aligned}
f^T D(\gamma P - I)f &= -f^T Df + \gamma f^T D(Pf) \\
&= -||f||_D^2 + \gamma \sum_s \mu(s)f(s) \sum_{s'} P(s'|s)f(s') \\
&= -(1 - \gamma)||f||_D^2 - \gamma||f||_{\text{Dir}}^2 \\
&= -\mathcal{N}(f).
\end{aligned}
$$

$\square$

The following lemmas are some variations of the mean-value theorem. In this lemma and below, we adopt the notation that gradients are row vectors.

**Lemma A.2.**   *(a). Let $h : \mathbb{R} \to \mathbb{R}$ be any differentiable function. For any $x, y \in \mathbb{R}$, there exists $\lambda \in (0, 1)$ and $z = \lambda x + (1 - \lambda)y$ such that*

$$h(y) - h(x) = h'(z)(y - x).$$

*(b). Let $\xi : \mathbb{R}^a \to \mathbb{R}$ be any differentiable function. For any $x, y \in \mathbb{R}^a$, there exists $\lambda \in (0, 1)$ and $z = \lambda x + (1 - \lambda)y$ such that*

$$\xi(y) - \xi(x) = \xi'(z)(y - x).$$

*(c). Let $f : \mathbb{R}^a \to \mathbb{R}^b$ be any differentiable function and $e \in \mathbb{R}^b$ be any vector. For any $x, y \in \mathbb{R}^a$, there exists $\lambda \in (0, 1)$ and $z = \lambda x + (1 - \lambda)y$ such that*

$$e^T(f(y) - f(x)) = e^T f'(z)(y - x),$$

*where $f'(z)$ is the Jacobian at $z$.*

*Proof.*   (a). This is a direct result of the well-known mean value theorem (Theorem 5.10 in Rudin (1976)).

(b). Define $h : \mathbb{R} \to \mathbb{R}$ such that $h(w) = \xi(x + w(y - x))$. Using the above fact, for any $u, v \in \mathbb{R}$, there exists $\lambda \in (0, 1)$ such that

$$h(v) - h(u) = h'(\lambda u + (1 - \lambda)v)(v - u).$$

By letting $v = 1$ and $u = 0$,

$$\xi(y) - \xi(x) = \xi'(\lambda x + (1 - \lambda)y)(y - x).$$

(c). Define $\xi : \mathbb{R}^a \to \mathbb{R}$ such that $\xi = e^T f$. Using the above fact, for any vectors $x, y \in \mathbb{R}^a$, there exists $\lambda \in (0, 1)$ and $z = \lambda x + (1 - \lambda)y$ such that

$$\xi(y) - \xi(x) = \xi'(z)(y - x).$$

Notice that in this case, $\xi(y) - \xi(x) = e^T(f(y) - f(x))$ and $\xi'(z) = e^T f'(z)$. So for any vectors $x, y$, there exists $z = \lambda x + (1 - \lambda)y$ such that

$$e^T f(y) - e^T f(x) = e^T f'(z)(y - x),$$

which is exactly what needs to be proved.

$\square$

**Lemma A.3.** *Let $f : \mathbb{R}^a \to \mathbb{R}^b$ be any differentiable function. For any $x, y \in \mathbb{R}^a$, there exists $\lambda \in (0, 1)$ and $z = \lambda x + (1 - \lambda)y$ such that*

$$||f(y) - f(x)|| \leq ||f'(z)|| \, ||y - x||.$$

*Proof.* Let us take $e = f(y) - f(x)$ in Lemma A.2. We thus have

$$(f(y) - f(x))^T (f(y) - f(x)) = (f(y) - f(x))^T f'(z)(y - x).$$

We now apply Cauchy-Schwarz inequality on the right hand side to obtain,

$$||f(y) - f(x)||^2 \leq ||f(y) - f(x)|| \cdot ||f'(z)(y - x)||,$$

and finally using the definition of matrix norm,

$$||f(y) - f(x)|| \leq ||f'(z)(y - x)|| \leq ||f'(z)|| \cdot ||y - x||.$$

$\square$

The next lemma shows how the mean-value theorem can help to analyze Neural TD Learning.

**Lemma A.4.** *The Projected Neural TD Learning with a mean-path update can be rewritten as*

$$\theta_{t+1} = \text{Proj}(\theta_t + \alpha_t(\bar{g}_1(\theta_t) + \bar{g}_2(\theta_t) + \bar{g}_3(\theta_t)))$$

*with $\bar{g}_1(\theta_t), \bar{g}_2(\theta_t), \bar{g}_3(\theta_t)$ defined as follows:*

$$\bar{g}_1(\theta_t) = \nabla_\theta V(\theta_1^{\text{mid}})^T D(\gamma P - I)(V(\theta_t) - V(\hat{\theta}_*)), \tag{10}$$

$$\bar{g}_2(\theta_t) = (\nabla_\theta V(\theta_t) - \nabla_\theta V(\theta_1^{\text{mid}}))^T D(\gamma P - I)(V(\theta_t) - V(\hat{\theta}_*)), \tag{11}$$

$$\bar{g}_3(\theta_t) = \nabla_\theta V(\theta_t)^T D(\gamma P - I)(\hat{V}^* - V^*), \tag{12}$$

*where $\lambda \in [0, 1]$ is a scalar and $\theta_1^{\text{mid}} = \lambda \theta_t + (1 - \lambda)\hat{\theta}_*$ is a vector such that*

$$(\theta_t - \hat{\theta}_*)^T \nabla_\theta V(\theta_1^{\text{mid}})^T D(\gamma P - I)(V(\theta_t) - V(\hat{\theta}_*)) = (V(\theta_t) - V(\hat{\theta}_*))^T D(\gamma P - I)(V(\theta_t) - V(\hat{\theta}_*)). \tag{13}$$

*Proof.* By Eq. (7),

$$\begin{aligned} \bar{g}(\theta_t) &= \nabla_\theta V(\theta_t)^T D(\gamma P - I)(V(\theta_t) - V^*) \\ &= \nabla_\theta V(\theta_t)^T D(\gamma P - I)(V(\theta_t) - V(\hat{\theta}_*)) + \nabla_\theta V(\theta_t)^T D(\gamma P - I)(\hat{V}^* - V^*). \end{aligned} \tag{14}$$

Now let $D(\gamma P - I)(V(\theta_t) - V(\hat{\theta}_*))$ be the vector $e$ in Lemma A.2. There exists a scalar $\lambda \in (0, 1)$ and a vector $\theta_1^{\mathrm{mid}} = \lambda\theta_t + (1 - \lambda)\hat{\theta}_*$ such that

$$(\theta_t - \hat{\theta}_*)^T \nabla_\theta V(\theta_1^{\mathrm{mid}})^T D(\gamma P - I)(V(\theta_t) - V(\hat{\theta}_*)) = (V(\theta_t) - V(\hat{\theta}_*))^T D(\gamma P - I)(V(\theta_t) - V(\hat{\theta}_*)).$$

This gives the reason to divide Eq.(14) into two parts as follows:

$$\begin{aligned}
&\nabla_\theta V(\theta_t)^T D(\gamma P - I)(V(\theta_t) - V(\hat{\theta}_*)) \\
=& \nabla_\theta V(\theta_1^{\mathrm{mid}})^T D(\gamma P - I)(V(\theta_t) - V(\hat{\theta}_*)) \\
& + (\nabla_\theta V(\theta_t) - \nabla_\theta V(\theta_1^{\mathrm{mid}}))^T D(\gamma P - I)(V(\theta_t) - V(\hat{\theta}_*)).
\end{aligned}$$

$\square$

The following lemma builds the relationship between $x^T D y$ and expectation.

**Lemma A.5.** *Let $x, y$ be two vectors and $x(i), y(i)$ denote their $i$'th entries, respectively. Let $\bar{x}, \bar{y}$ be two scalars such that $|x(i)| \leq \bar{x}$ and $|y(i)| \leq \bar{y}$ hold for all $i$. The following results hold:*

$$x^T D y = y^T D x \leq \bar{x}\bar{y},$$

$$x^T D P y \leq \bar{x}\bar{y},$$

$$y^T D P x \leq \bar{x}\bar{y}.$$

*Proof.* We expand $x^T D y$, $x^T D P y$ and $y^T D P x$ as follows:

$$x^T D y = y^T D x = \sum_i \mu(s_i) x(i) y(i) \leq \sum_i \mu(s_i) \bar{x}\bar{y} \leq \bar{x}\bar{y},$$

$$x^T D P y = \sum_i \mu(s_i) x(i) \sum_j P(s_j|s_i) y(j) \leq \bar{x}\bar{y} \sum_i \mu(s_i) \sum_j P(s_j|s_i) \leq \bar{x}\bar{y},$$

$$y^T D P x = \sum_i \mu(s_i) y(i) \sum_j P(s_j|s_i) x(j) \leq \bar{x}\bar{y} \sum_j \sum_i \mu(s_i) P(s_j|s_i) \leq \bar{x}\bar{y}.$$

$\square$

The following lemmas implies two important properties of neural network approximation: Lipschitzness and smoothness.

**Lemma A.6.** *For all $k \in \{1, 2, \ldots, K\}$,*

$$||\theta^{(k)}|| \leq O(\sqrt{m}).$$

*Proof.*

$$\begin{aligned}
||\theta^{(k)}|| &\leq ||\theta^{(k)} - \theta_0^{(k)}|| + ||\theta_0^{(k)}|| \\
&\leq \omega + ||\theta_0^{(k)}|| \\
&\leq O(\sqrt{m}),
\end{aligned}$$

where the second inequality is by projection and the last inequality uses Assumption 3.1 and the fact $\omega$ is constant to $m$. $\square$

**Lemma A.7.** *For all $k \in \{1, 2, \ldots, K\}$,*

$$||x^{(k)}|| \leq O(\sqrt{m}).$$

*Proof.* From Assumption 2.3, $||x^{(0)}|| \le 1$. By Lemma A.6,

$$\begin{aligned}
||x^{(1)}||^2 = \left|\left|\frac{1}{\sqrt{m}}\sigma(\theta^{(1)}x^{(0)})\right|\right|^2 \\
\le \frac{1}{m}l^2||\theta^{(1)}||^2||x^{(0)}||^2 + |\sigma(0)|^2 \\
\le O(m).
\end{aligned}$$

By induction, suppose $||x^{(k)}||^2 \le O(m)$. By Lemma A.6,

$$\begin{aligned}
||x^{(k+1)}||^2 = ||\frac{1}{\sqrt{m}}\sigma(\theta^{(k+1)}x^{(k)})||^2 \\
\le \frac{1}{m}l^2||\theta^{(k+1)}||^2||x^{(k)}||^2 + |\sigma(0)|^2 \\
\le O(m).
\end{aligned}$$

$\square$

**Lemma A.8.** *For all $k \in \{1, 2, \dots, K\}$,*

$$||\nabla_{x^{(k-1)}}x^{(k)}|| \le O(1).$$

*Proof.*

$$\nabla_{x^{(k-1)}}x^{(k)}(i,j) = \frac{1}{\sqrt{m}}\sigma'\left(\sum_j \theta^{(k)}(i,j)x^{(k-1)}(j)\right)\theta^{(k)}(i,j),$$

which implies

$$\begin{aligned}
||\nabla_{x^{(k-1)}}x^{(k)}||^2 &= \sup_{||v||=1} \sum_{i=1}^m \left(\sum_j \nabla_{x^{(k-1)}}x^{(k)}(i,j)v_j\right)^2 \\
&= \sup_{||v||=1} \frac{1}{m}||\Sigma'\theta^{(k)}v||^2 \\
&\le \frac{1}{m}||\Sigma'||^2 \cdot ||\theta^{(k)}||^2 \\
&\le O(1).
\end{aligned}$$

where $\Sigma'$ is a diagonal matrix with $\Sigma'(i,i) = \sigma'(\sum_j \theta^{(k)}(i,j)x^{(k-1)}(j))$. The last inequality holds because of Lemma A.6. $\square$

**Lemma A.9.** *For all $k \in \{1, 2, \dots, K\}$,*

$$||\nabla_{\theta^{(k)}}x^{(k)}|| \le O(1)$$

*Here, $\nabla_{\theta^{(k)}}x^{(k)}$ is defined to be a matrix whose $(i, (j-1)m+h)$'th entry $\nabla_{\theta^{(k)}}x^{(k)}(i,j,h)$ is given by*

$$\nabla_{\theta^{(k)}}x^{(k)}(i,j,h) = \frac{\partial x^{(k)}(i)}{\partial \theta^{(k)}(j,h)}.$$

*Proof.*

$$\nabla_{\theta^{(k)}}x^{(k)}(i,j,j') = \frac{1}{\sqrt{m}}\mathbf{1}\{i=j\}\sigma'\left(\sum_h \theta^{(k)}(i,h)x^{(k-1)}(h)\right)x^{(k-1)}(j').$$

We can write this as

$$\nabla_{\theta^{(k)}}x^{(k)}(i,j,j') = \frac{1}{\sqrt{m}}\mathbf{1}\{i=j\}\xi(i)x^{(k-1)}(j')$$

which implies

$$
\begin{aligned}
||\nabla_{\theta^{(k)}} x^{(k)}||^2 &= \sup_{||V||_F=1} \sum_{i=1}^{m} \left( \sum_{j,j'} \nabla_{\theta^{(k)}} x^{(k)}(i,j,j') V_{j,j'} \right)^2 \\
&= \frac{1}{m} \sup_{||V||_F=1} \sum_{i=1}^{m} \left( \sum_{j,j'} \mathbf{1}\{i=j\}\xi(i) x^{(k-1)}(j') V_{j,j'} \right)^2 \\
&= \frac{1}{m} \sup_{||V||_F=1} \sum_{i=1}^{m} \left( \sum_{j} \mathbf{1}\{i=j\}\xi(i) [Vx^{k-1}]_j \right)^2 \\
&= \frac{1}{m} \sup_{||V||_F=1} \sum_{i=1}^{m} \xi(i)^2 \left[ Vx^{k-1} \right]_i^2 \\
&= \sup_{||V||_F=1} \frac{1}{m} ||\Sigma' V x^{(k-1)}||^2 \\
&\leq \frac{1}{m} ||\Sigma'||^2 \cdot ||x^{(k-1)}||^2 \\
&\leq O(1).
\end{aligned}
$$

The last inequality holds because of Lemma A.7. $\qquad\square$

**Lemma A.10.** *For all $s \in S$ and $\theta$,*

$$||\nabla_\theta V(s,\theta)|| \leq O(1).$$

*with respect to $m$.*

*Proof.* Since each entry of $b$ satisfies $|b_r| \leq 1$, it is easy to see that

$$||\nabla_{x^{(K)}} V(s,\theta)|| = \frac{1}{\sqrt{m}} ||b|| \leq 1.$$

By Lemma A.8, Lemma A.9, and the chain rule,

$$||\nabla_{\theta^{(k)}} V(s,\theta)|| = ||\nabla_{x^{(K)}} V(s,\theta) \nabla_{x^{(K-1)}} x^{(K)} \cdots \nabla_{x^{(k)}} x^{(k+1)} \nabla_{\theta^{(k)}} x^{(k)}|| \leq O(1).$$

It follows:

$$||\nabla_\theta V(s,\theta)||^2 = \sup_{||V||_F=1} \sum_{k=1}^{K} \left( \nabla_{\theta^{(k)}} V(s,\theta) V_k \right)^2 \leq O(1).$$

$\qquad\square$

**Lemma A.11.** *The Hessian matrix, which is $\nabla_\theta^2 V(s,\theta)$, has a norm of $O(m^{-0.5})$. In other words,*

$$||\nabla_\theta^2 V(s,\theta)|| \leq O(m^{-0.5}).$$

This is a direct result of Theorem 3.2 in Liu et al. (2020). Notice that since we assume Assumption 3.1 and 2.5 (which correspond to Lemma G.1 and Lemma G.4 in Liu et al. (2020)) holds with probability 1, Lemma A.11 also holds with probability 1.

The following lemma implies that the update during each step will not be too large.

**Lemma A.12.** *We have the following bound for $||g(\theta_t)||$:*

$$||g(\theta_t)||^2 \leq O(\epsilon^2) + \frac{1}{(1-\gamma)^2} O(1).$$

*Proof.* Recall that Eq. (1) tell us the property $V^*$ satisfies, and it can be rewritten as

$$V^*(s) = r(s) + \gamma \sum_{s''} P(s''|s) V^*(s'').$$

Moreover, recall that $g(\theta_t)$ is defined to be

$$g(\theta_t) = \nabla_\theta V(s, \theta_t) \left[ r(s) + \gamma V(s', \theta_t) - V(s, \theta_t) \right]$$

. This immediately implies that $g(\theta_t)$ is actually a random variable and implicitly relies on the state $s$. This allows $g(\theta_t)$ to be rewritten as

$$
\begin{aligned}
g(\theta_t) =& \nabla_\theta V(s, \theta_t) \left[ r(s) + \gamma V(s', \theta_t) - V(s, \theta_t) \right] \\
=& \nabla_\theta V(s, \theta_t) \left[ V^*(s) - V(s, \theta_t) + \gamma \sum_{s''} P(s''|s)(V(s', \theta_t) - V^*(s'')) \right] \\
=& \nabla_\theta V(s, \theta_t) \left[ \left( V(s, \hat{\theta}_*) - V(s, \theta_t) \right) + \left( V^*(s) - V(s, \hat{\theta}_*) \right) \right. \\
& \left. + \gamma \sum_{s''} P(s''|s) \left( V(s', \theta_t) - V(s', \hat{\theta}_*) + V(s', \hat{\theta}_*) - V^*(s') + V^*(s') - V^*(s'') \right) \right] \\
=& \nabla_\theta V(s, \theta_t) \left[ \hat{f}(s) + \left( V^*(s) - V(s, \hat{\theta}_*) \right) \right. \\
& \left. + \gamma \sum_{s''} P(s''|s) \left( -\hat{f}(s') + V(s', \hat{\theta}_*) - V^*(s') + V^*(s') - V^*(s'') \right) \right].
\end{aligned}
$$

where, for simplicity, we denote $\hat{f}(s) = V(s, \hat{\theta}_*) - V(s, \theta_t)$. Now, $||g(\theta_t)||^2$ can be bounded as

$$
\begin{aligned}
||g(\theta_t)||^2 \leq & 5 ||\nabla_\theta V(s, \theta_t)||^2 \left[ \hat{f}(s)^2 + \left( \gamma \sum_{s''} P(s''|s) \hat{f}(s') \right)^2 + \left( V^*(s) - V(s, \hat{\theta}_*) \right)^2 \right. \\
& \left. + \left( \gamma \sum_{s''} P(s''|s)(V(s', \hat{\theta}_*) - V^*(s')) \right)^2 + \left( \gamma \sum_{s''} P(s''|s)(V^*(s') - V^*(s'')) \right)^2 \right].
\end{aligned}
$$

There are five different terms, and now we will bound them respectively. By Lemma A.10 which says $\hat{f}(s)$ is $O(1)$-Lipschitz and the fact that $||\theta_t - \hat{\theta}_*|| \leq \omega$,

$$\hat{f}(s)^2 \leq O(1).$$

Using the above fact,

$$\left[ \gamma \sum_{s''} P(s''|s) \hat{f}(s') \right]^2 = \gamma^2 \hat{f}(s')^2 \leq \gamma^2 O(1).$$

Using Eq.(8) , we can derive

$$\left[ V^*(s) - V(s, \hat{\theta}_*) \right]^2 \leq O(\epsilon^2).$$

Using the above fact,

$$\left[ \gamma \sum_{s''} P(s''|s)(V(s', \hat{\theta}_*) - V^*(s')) \right]^2 = \gamma^2 \left[ V(s', \hat{\theta}_*) - V^*(s') \right]^2 \leq \gamma^2 O(\epsilon^2).$$

Using Eq.(2) and Jensen's inequality,

$$\left[ \gamma \sum_{s''} P(s''|s)(V^*(s') - V^*(s'')) \right] \leq \gamma^2 \sum_{s''} P(s''|s) \left( \frac{2 r_{\max}}{1 - \gamma} \right)^2 = \frac{\gamma^2}{(1 - \gamma)^2} O(1).$$

Combine the above five facts, and we have the bound for $||g(\theta_t)||^2$:

$$
\begin{aligned}
||g(\theta_t)||^2 \leq & 5||\nabla_\theta V(s, \theta_t)||^2 \Bigg[\hat{f}(s)^2 + \left(\gamma \sum_{s''} P(s''|s)\hat{f}(s')\right)^2 + \left(V^*(s) - V(s, \hat{\theta}_*)\right)^2 \\
& + \left(\gamma \sum_{s''} P(s''|s)(V(s', \hat{\theta}_*) - V^*(s'))\right)^2 + \left(\gamma \sum_{s''} P(s''|s)(V^*(s') - V^*(s''))\right)^2 \Bigg] \\
\leq & 5||\nabla_\theta V(s, \theta_t)||^2 \left(O(1) + \gamma^2 O(1) + O(\epsilon^2) + \gamma^2 O(\epsilon^2) + \frac{\gamma^2}{(1-\gamma)^2} O(1)\right) \\
\leq & (1 + \gamma^2)O(1 + \epsilon^2) + \frac{\gamma^2}{(1-\gamma)^2} O(1)
\end{aligned}
\tag{15}
$$

where the last inequality uses Lemma A.10. Since $0 \leq \gamma \leq 1$, we can simplify the bound as

$$
||g(\theta_t)||^2 \leq O(\epsilon^2) + \frac{1}{(1-\gamma)^2} O(1).
$$

$\square$

## A.2 PROOF FOR THE MAIN RESULT

*Proof.* Consider Projected Neural TD Learning, it is easy to see

$$
\begin{aligned}
||\theta_{t+1} - \hat{\theta}_*||^2 &= ||\mathbf{Proj}(\theta_t + \alpha_t g(\theta_t)) - \hat{\theta}_*||^2 \\
&\leq ||\theta_t - \hat{\theta}_* + \alpha_t g(\theta_t)||^2 \\
&= ||\theta_t - \hat{\theta}_*||^2 + 2\alpha_t(\theta_t - \hat{\theta}_*)^T g(\theta_t) + \alpha_t^2 ||g(\theta_t)||^2 \\
&= ||\theta_t - \hat{\theta}_*||^2 + 2\alpha_t(\theta_t - \hat{\theta}_*)^T \bar{g}(\theta_t) + \alpha_t^2 ||g(\theta_t)||^2 + 2\alpha_t(\theta_t - \hat{\theta}_*)^T (g(\theta_t) - \bar{g}(\theta_t)).
\end{aligned}
\tag{16}
$$

First, we consider $2\alpha_t(\theta_t - \hat{\theta}_*)^T \bar{g}(\theta_t)$. Lemma A.4 allows us to divide it into several parts and bound them respectively as follows:

$$
2\alpha_t(\theta_t - \hat{\theta}_*)^T \bar{g}(\theta_t) = 2\alpha_t(\theta_t - \hat{\theta}_*)^T (\bar{g}_1(\theta_t) + \bar{g}_2(\theta_t) + \bar{g}_3(\theta_t)).
$$

To bound $(\theta_t - \hat{\theta}_*)^T \bar{g}_1(\theta_t)$,

$$
\begin{aligned}
& (\theta_t - \hat{\theta}_*)^T \bar{g}_1(\theta_t) \\
=& (\theta_t - \hat{\theta}_*)^T \nabla_\theta V(\theta_1^{\mathrm{mid}})^T D(\gamma P - I)(V(\theta_t) - V(\hat{\theta}_*)) \\
=& (V(\theta_t) - V(\hat{\theta}_*))^T D(\gamma P - I)(V(\theta_t) - V(\hat{\theta}_*)) \\
=& -\mathcal{N}(V(\theta_t) - V(\hat{\theta}_*)),
\end{aligned}
$$

where the first equality is by Eq.(10), the second equality is by Eq. (13), and the third equality is by setting $f = V(\theta_t) - V(\hat{\theta}_*)$ in Lemma A.1.

To bound $(\theta_t - \hat{\theta}_*)^T \bar{g}_2(\theta_t)$, we first notice that Lemma A.11 means $V(s, \theta)$ is $O(m^{-0.5})$-smoothness with respect to $\theta$. Hence,

$$
||\nabla_\theta V(s, \theta_t) - \nabla_\theta V(s, \theta_1^{\mathrm{mid}})|| \leq O(m^{-0.5}) \cdot ||\theta_t - \theta_1^{\mathrm{mid}}|| = O(m^{-0.5}),
$$

where the inequality is by Lemma A.11 and the equality is because both $\theta_t$ and $\theta_1^{\mathrm{mid}}$ are in $B(\theta_0, \omega)$. Similarly, Lemma A.10 tells us that $V(s, \theta)$ is $O(1)$-Lipschitz. This means

$$
||V(s, \theta_t) - V(s, \hat{\theta}_*)|| \leq O(1) \cdot ||\theta_t - \hat{\theta}_*|| = O(1),
$$

where the inequality is by Lemma A.10 and the equality is because both $\theta_t$ and $\hat{\theta}_*$ lie in $B(\theta_0.\omega)$. These two facts imply that each entry of $(\nabla_\theta V(\theta_t) - \nabla_\theta V(\theta_1^{\mathrm{mid}}))(\theta_t - \hat{\theta}_*)$ is upper-bounded by

$O(m^{-0.5})$ and each entry of $V(\theta_t) - V(\hat{\theta}_*)$ is upper-bounded by $O(1)$. With this fact,

$$
\begin{aligned}
&(\theta_t - \hat{\theta}_*)^T \bar{g}_2(\theta_t) \\
=&(\theta_t - \hat{\theta}_*)^T (\nabla_\theta V(\theta_t) - \nabla_\theta V(\theta_1^{\mathrm{mid}}))^T D(\gamma P - I)(V(\theta_t) - V(\hat{\theta}_*)) \\
\leq&(1 + \gamma)O(m^{-0.5}) \\
\leq&O(m^{-0.5}).
\end{aligned}
$$

where the equality is by Eq.(11) and the first inequality is by setting $x$ to be $(\nabla_\theta V(\theta_t) - \nabla_\theta V(\theta_1^{\mathrm{mid}}))(\theta_t - \hat{\theta}_*)$, $y$ to be $V(\theta_t) - V(\hat{\theta}_*)$ in Lemma A.5.

To bound $(\theta_t - \hat{\theta}_*)^T \bar{g}_3(\theta_t)$,

$$
\begin{aligned}
&(\theta_t - \hat{\theta}_*)^T \bar{g}_3(\theta_t) \\
=&(\theta_t - \hat{\theta}_*)^T \nabla_\theta V(\theta_t)^T D(\gamma P - I)(\hat{V}^* - V^*) \\
\leq&(1 + \gamma)O(\epsilon) \\
\leq&O(\epsilon),
\end{aligned}
$$

where the equality is by Eq.(12) and the first inequality is by setting $x$ to be $\nabla_\theta V(\theta_t)(\theta_t - \hat{\theta}_*)$, $y$ to be $\hat{V}^* - V^*$ in Lemma A.5, as each entry of $\nabla_\theta V(\theta_t)(\theta_t - \hat{\theta}_*)$ is bounded by $O(1)$ using Lemma A.10 and each entry of $\hat{V}^* - V^*$ is bounded by $\epsilon$ using Assumption 8.

Combining the above facts,

$$
2\alpha_t(\theta_t - \hat{\theta}_*)^T \bar{g}(\theta_t) \leq -2\alpha_t \mathcal{N}(V(\theta_t) - V(\hat{\theta}_*)) + O(\alpha_t(m^{-0.5} + \epsilon)),
$$

which is the bound of the first part in Eq.(16).

Second, we consider $\alpha_t^2 \|g(\theta_t)\|^2$. By Lemma A.12,

$$
\alpha_t^2 \|g(\theta_t)\|^2 \leq O(\alpha_t^2 \epsilon^2) + \frac{1}{(1 - \gamma)^2} O(\alpha_t^2).
$$

The above facts are all we need to establish the result when each $s_t$ is sampled i.i.d. from $\mu$. In summary,

$$
\begin{aligned}
2\alpha_t(\theta_t - \hat{\theta}_*)^T \bar{g}(\theta_t) + \alpha_t^2 \|g(\theta_t)\|^2 \leq& -2\alpha_t \mathcal{N}(V(\theta_t) - V(\hat{\theta}_*)) \\
&+ O(\alpha_t(m^{-0.5} + \epsilon)) + O(\alpha_t^2 \epsilon^2) + \frac{1}{(1 - \gamma)^2} O(\alpha_t^2),
\end{aligned}
$$

which just simply combines the bound for $2\alpha_t(\theta_t - \hat{\theta}_*)^T \bar{g}(\theta_t)$ and $\alpha_t^2 \|g(\theta_t)\|^2$ we obtained earlier.

Given $\theta_t$, taking expectation on both sides of Eq.(16):

$$
\begin{aligned}
&\mathbb{E}[\|\theta_{t+1} - \hat{\theta}_t\|^2] \\
\leq&\|\theta_t - \hat{\theta}_*\|^2 + 2\alpha_t(\theta_t - \hat{\theta}_*)\bar{g}(\theta_t) + \mathbb{E}[\alpha_t^2 \|g(\theta_t)\|^2] + 2\alpha_t(\theta_t - \hat{\theta}_*)\mathbb{E}[g(\theta_t) - \bar{g}(\theta_t)] \\
=&\|\theta_t - \hat{\theta}_*\|^2 + 2\alpha_t(\theta_t - \hat{\theta}_*)\bar{g}(\theta_t) + \mathbb{E}[\alpha_t^2 \|g(\theta_t)\|^2] \\
\leq&\|\theta_t - \hat{\theta}_*\|^2 - 2\alpha_t \mathcal{N}(V(\theta_t) - V(\hat{\theta}_*)) + O(\alpha_t(m^{-0.5} + \epsilon)) + O(\alpha_t^2 \epsilon^2) + \frac{1}{(1 - \gamma)^2} O(\alpha_t^2),
\end{aligned}
$$

where the equality uses the condition that given $s_t$ are sampled i.i.d. from $\mu$, which will lead to $\mathbb{E}[g(\theta_t) - \bar{g}(\theta_t)] = 0$. From now on we fix $\alpha_t = \alpha$, and this leads to

$$
\frac{\mathbb{E}[\|\theta_{t+1} - \hat{\theta}_*\|^2]}{2\alpha} - \frac{\mathbb{E}[\|\theta_t - \hat{\theta}_*\|^2]}{2\alpha} \leq -\mathbb{E}[\mathcal{N}(V(\theta_t) - V(\hat{\theta}_*))] + O(m^{-0.5} + \epsilon) + O(\alpha \epsilon^2) + \frac{1}{(1 - \gamma)^2} O(\alpha).
$$

Telescoping a sum from 1 to $T$ and dividing both sides by $T$, we establish the result in i.i.d. case.

To continue with the non i.i.d. case, we need to consider $2\alpha(\theta_t - \hat{\theta}_*)^T(g(\theta_t) - \bar{g}(\theta_t))$ in Eq.(16) which is no longer 0. We will use $\tau_{\mathrm{mix}}$, the mixing time defined in Eq.(3), to split it into two terms and bound them separately. This idea is from Sun et al. (2018).

Our first step is to divide it into two terms:

$$2\alpha(\theta_t - \hat{\theta}_*)(g(\theta_t) - \bar{g}(\theta_t)) = 2\alpha(\theta_t - \theta_{t-\tau_{\mathrm{mix}}})(g(\theta_t) - \bar{g}(\theta_t)) + 2\alpha(\theta_{t-\tau_{\mathrm{mix}}} - \hat{\theta}_*)(g(\theta_t) - \bar{g}(\theta_t)), \tag{17}$$

To bound $2\alpha(\theta_t - \theta_{t-\tau_{\mathrm{mix}}})(g(\theta_t) - \bar{g}(\theta_t))$, notice that Lemma A.12 implies $||g(\theta_t)|| \leq O(\epsilon) + \frac{1}{1-\gamma}O(1)$. As a consequence of Eq.(6), we have

$$\begin{aligned}
||\theta_t - \theta_{t-\tau_{\mathrm{mix}}}|| &\leq ||\theta_{t-1} - \theta_{t-\tau_{\mathrm{mix}}} + \alpha g(\theta_{t-1})|| \\
&\leq ||\theta_{t-2} - \theta_{t-\tau_{\mathrm{mix}}} + \alpha\left[g(\theta_{t-1}) + g(\theta_{t-2})\right]|| \\
&\leq \cdots \\
&\leq ||\theta_{t-\tau_{\mathrm{mix}}} - \theta_{t-\tau_{\mathrm{mix}}} + \alpha\left[g(\theta_{t-1}) + g(\theta_{t-2}) + \cdots + g(\theta_{t-\tau_{\mathrm{mix}}})\right]|| \\
&\leq \left[O(\epsilon) + \frac{1}{1-\gamma}O(1)\right]\alpha\tau_{\mathrm{mix}}.
\end{aligned}$$

Further, the same upper bound that $g(\theta_t)$ has holds for $\bar{g}(\theta_t)$ since $\bar{g}(\theta_t) = \mathbb{E}[g(\theta_t)]$. These observations imply that

$$2\alpha(\theta_t - \theta_{t-\tau_{\mathrm{mix}}})(g(\theta_t) - \bar{g}(\theta_t)) \leq \alpha^2\tau_{\mathrm{mix}}\left[O(\epsilon^2) + \frac{1}{(1-\gamma)^2}O(1)\right].$$

To bound $2\alpha(\theta_{t-\tau_{\mathrm{mix}}} - \hat{\theta}_*)(g(\theta_t) - \bar{g}(\theta_t))$, observe that, conditioned on $\theta_{t-\tau_{\mathrm{mix}}}$, the quantity $\theta_{t-\tau_{\mathrm{mix}}} - \hat{\theta}_*$ is deterministic, and $\mathbb{E}[g(\theta_t) - \bar{g}(\theta_t)|\theta_{t-\tau\mathrm{mix}}]$ can be bounded by

$$\begin{aligned}
\mathbb{E}[g(\theta_t) - \bar{g}(\theta_t)|\theta_{t-\tau_{\mathrm{mix}}}] &\leq \left[O(\epsilon) + \frac{1}{1-\gamma}O(1)\right]\max_s ||P^{\tau_{\mathrm{mix}}}_{s,:} - \mu||_{\mathrm{TV}} \\
&\leq \left[O(\epsilon) + \frac{1}{1-\gamma}O(1)\right]\epsilon_{\mathrm{mix}}.
\end{aligned}$$

The first inequality follows because, conditional on $s_{t-\tau_{\mathrm{mix}}}$, the random variable $s_t$ has the distribution of one row of $P^{\tau_{\mathrm{mix}}}$. Thus, the second term in Eq.(17) can be bounded as

$$\begin{aligned}
\mathbb{E}[2\alpha(\theta_{t-\tau_{\mathrm{mix}}} - \hat{\theta}_*)(g(\theta_t) - \bar{g}(\theta_t))] &= \mathbb{E}[\mathbb{E}[2\alpha(\theta_{t-\tau_{\mathrm{mix}}} - \hat{\theta}_*)(g(\theta_t) - \bar{g}(\theta_t))|\theta_{t-\tau_{\mathrm{mix}}}]] \\
&\leq \left[O(\epsilon) + \frac{1}{1-\gamma}O(1)\right]\alpha\epsilon_{\mathrm{mix}}.
\end{aligned}$$

Thus, coming back to Eq.(17), we obtain that

$$\mathbb{E}[2\alpha(\theta_t - \hat{\theta}_*)(g(\theta_t) - \bar{g}(\theta_t))] \leq \alpha^2\tau_{\mathrm{mix}}\left[O(\epsilon^2) + \frac{1}{(1-\gamma)^2}O(1)\right] + \left[O(\epsilon) + \frac{1}{1-\gamma}O(1)\right]\alpha\epsilon_{\mathrm{mix}}.$$

Now let $\epsilon_{\mathrm{mix}} = \alpha$, by the definition of $\tau_{\mathrm{mix}}$, $\tau_{\mathrm{mix}} = \frac{\log\frac{\alpha}{C}}{\log\beta}$. Using the fact that $\log x \leq x-1, \forall x > 0$, we derive

$$\frac{\log\frac{\alpha}{C}}{\log\beta} \leq \frac{\log\frac{\alpha}{C}}{\beta - 1} = \frac{\log\frac{C}{\alpha}}{1 - \beta},$$

where the first inequality is because $\log\beta \leq \beta - 1 < 0$ and $\log\frac{\alpha}{C} \leq 0$. The bound can be rewritten as

$$\mathbb{E}[2\alpha_t(\theta_t - \hat{\theta}_*)(g(\theta_t) - \bar{g}(\theta_t))] \leq \alpha^2\frac{\log\frac{C}{\alpha}}{1 - \beta}\left[O(\epsilon^2) + \frac{1}{(1-\gamma)^2}O(1)\right] + \left[O(\epsilon) + \frac{1}{1-\gamma}O(1)\right]\alpha^2.$$

Now we are ready to consider Eq.(16). Taking expectation on both sides,

$$\begin{aligned}
&\mathbb{E}[2\alpha(\theta_t - \hat{\theta}_*)^T g(\theta_t) + \alpha^2||g(\theta_t)||^2] \\
&\leq -2\alpha\mathcal{N}(V(\theta_t) - V(\hat{\theta}_*)) + O(\alpha(m^{-0.5} + \epsilon)) + O(\alpha^2\epsilon^2) + \frac{1}{(1-\gamma)^2}O(\alpha^2) \\
&\quad + \alpha^2\frac{\log\frac{C}{\alpha}}{1 - \beta}\left[O(\epsilon^2) + \frac{1}{(1-\gamma)^2}O(1)\right] + \left[O(\epsilon) + \frac{1}{1-\gamma}O(1)\right]\alpha^2 \\
&= -2\alpha\mathcal{N}(V(\theta_t) - V(\hat{\theta}_*)) + O(\alpha(m^{-0.5} + \epsilon)) + O(\alpha^2\epsilon^2) + \frac{1}{(1-\gamma)^2}O(\alpha^2) \\
&\quad + O(\alpha^2\frac{\log\frac{C}{\alpha}}{1 - \beta}\epsilon^2) + \frac{1}{(1-\gamma)^2}O(\alpha^2\frac{\log\frac{C}{\alpha}}{1 - \beta}).
\end{aligned}$$

Rewrite this inequality as

$$\frac{\mathbb{E}[||\theta_{t+1} - \hat{\theta}_*||^2]}{2\alpha} - \frac{\mathbb{E}[||\theta_t - \hat{\theta}_*||^2]}{2\alpha}$$
$$= -\mathcal{N}(V(\theta_t) - V(\hat{\theta}_*)) + O(m^{-0.5} + \epsilon) + O(\alpha\epsilon^2) + \frac{1}{(1-\gamma)^2}O(\alpha)$$
$$+ O(\alpha\frac{\log\frac{C}{\alpha}}{1-\beta}\epsilon^2) + \frac{1}{(1-\gamma)^2}O(\alpha\frac{\log\frac{C}{\alpha}}{1-\beta}).$$

Telescoping a sum from $1$ to $T$ and dividing both sides by $T$, we establish the result in non the i.i.d. case. $\qquad\square$

# B ANALYSIS OF UNPROJECTED TD LEARNING – PROOF OF OUR CONJECTURE FOR THE SINGLE HIDDEN LAYER CASE

## B.1 RESULT WITHOUT PROJECTION

Here we prove that when the distance from the optimal solution is $O(\sqrt{m})$, unprojected TD learning converges with high probability. We begin by standardizing notation.

Recall $g(\theta_t)$ is defined by

$$g(\theta_t) = \nabla_\theta V(s, \theta_t)\delta_t,$$

where

$$\delta_t = r(s) + \gamma V(s', \theta_t) - V(s, \theta_t).$$

Without projection, there is only one step in the algorithm, which is

$$\theta_{t+1} = \theta_t + \alpha_t g(\theta_t).$$

We call this algorithm Non Projected Neural TD Learning. Similarly to the way we proceeded earlier in this paper, Non Projected Neural TD Learning with *mean-path update* is given by

$$\theta_{t+1} = \theta_t + \alpha_t \bar{g}(\theta_t).$$

We will need to make the following assumption on exact approximation, which is stronger than what we assumed in the projected case.

**Assumption B.1.** *There exists some $\theta_*$ such that $V(\theta_*) = V^*$.*

Now, for simplicity of notations, we also introduce the following notation. Remember the smallest eigenvalue $\lambda_{\min}$ of some matrix $A$ is defined as

$$\lambda_{\min}(A) = \arg\min_x \frac{||Ax||}{||x||}.$$

A similar way can be applied to define the smallest $2, D$ eigenvalue $\sigma_{\min}^{2,D}$ of a matrix $A$:

$$\sigma_{\min}^{2,D}(A) = \arg\min_x \frac{||Ax||_D}{||x||}.$$

For simplicity, we will use $\sigma_{\min}^{2,D}$ to denote the $2, D$ eigenvalue of $\nabla_\theta V(\theta_*)$.

We will always suppose each $s_t$ is sampled i.i.d. from $\mu$. We now introduce the following result.

**Theorem B.1.** *In the Non Projected Neural TD learning using a one hidden layer neural network, smooth and Lipschitz activation function, suppose each $s_t$ are sampled i.i.d from $\mu$, and stepsize is chosen to be $\alpha_t = \frac{1}{\lambda(t+1)}$. Further, let $A$ be the event*

$$A = \left\{ \sup_t X_t < \frac{2(1-\gamma)(\sigma_{\min}^{2,D})^2\lambda - 3l^4(1+\gamma^2)}{4(1+\gamma)c_0 l}m^{0.5}, \right\}$$

*and assume*

$$\mathbb{E}\left[||\theta_0 - \theta_*||^2\right] < \frac{\left(2(1-\gamma)(\sigma_{\min}^{2,D})^2\lambda - 3l^4(1+\gamma^2)\right)^2}{64(1+\gamma)^2 c_0^2 l^2 \lambda^2}m\delta - 2.$$

*Then, $A$ happens with probability at least $1 - \delta$ and the sequence $\left\{\mathbb{E}\left[||\theta_t - \theta_*||^2|A\right]\right\}$ converges to 0.*

## B.2 USEFUL LEMMAS

**Lemma B.1.** $(\theta_t - \theta_*)^T \bar{g}(\theta_t)$ *can be rewritten as*

$$(\theta_t - \theta_*)^T \bar{g}(\theta_t) = \bar{g}_1(\theta_t) + \bar{g}_2(\theta_t) + \bar{g}_3(\theta_t),$$

*where $\bar{g}_1(\theta_t), \bar{g}_2(\theta_t), \bar{g}_3(\theta_t)$ are defined as follows:*

$$\bar{g}_1(\theta_t) = (\theta_t - \theta_*)^T \nabla_\theta V(\theta_*)^T D(\gamma P - I)\nabla_\theta V(\theta_*)(\theta_t - \theta_*), \tag{18}$$

$$\bar{g}_2(\theta_t) = (\theta_t - \theta_*)^T (\nabla_\theta V(\theta_t) - \nabla_\theta V(\theta_*))^T D(\gamma P - I)(V(\theta_t) - V(\theta_*)), \qquad (19)$$

$$\bar{g}_3(\theta_t) = (\theta_t - \theta_*)^T \nabla_\theta V(\theta_*)^T D(\gamma P - I)(\nabla_\theta V(\theta_2^{\mathrm{mid}}) - \nabla_\theta V(\theta_*))(\theta_t - \theta_*). \qquad (20)$$

*Here, $\lambda_2 \in [0,1]$ is a scalar and $\theta_2^{\mathrm{mid}} = \lambda_2 \theta_t + (1 - \lambda_2)\theta_*$ is a vector such that*

$$(\theta_t - \theta_*)^T \nabla_\theta V(\theta_*)^T D(\gamma P - I)\nabla_\theta V(\theta_4^{\mathrm{mid}})(\theta_t - \theta_*) = (\theta_t - \theta_*)^T \nabla_\theta V(\theta_*)^T D(\gamma P - I)(V(\theta_t) - V(\theta_*)).$$

*Proof.* First, we divide $(\theta_t - \theta_*)^T \bar{g}(\theta_t)$ into two parts,

$$
\begin{aligned}
(\theta_t - \theta_*)^T \bar{g}(\theta_t) &= (\theta_t - \theta_*)^T \nabla_\theta V(\theta_t)^T D(\gamma P - I)(V(\theta_t) - V(\theta_*)) \\
&= (\theta_t - \theta_*)^T \nabla_\theta V(\theta_*)^T D(\gamma P - I)(V(\theta_t) - V(\theta_*)) \\
&\quad + (\theta_t - \theta_*)^T (\nabla_\theta V(\theta_t) - \nabla_\theta V(\theta_*))^T D(\gamma P - I)(V(\theta_t) - V(\theta_*)) \\
&= (\theta_t - \theta_*)^T \nabla_\theta V(\theta_*)^T D(\gamma P - I)(V(\theta_t) - V(\theta_*)) + \bar{g}_2(\theta_t)
\end{aligned}
$$

where the first equality is by the definition of $\bar{g}(\theta_t)$ in Eq.(7). Now let $(\theta_t - \theta_*)^T \nabla_\theta V(\theta_*)^T D(\gamma P - I)$ be the vector $e$ in Lemma A.2. There exists a scalar $\lambda_2 \in [0,1]$ and a vector $\theta_2^{\mathrm{mid}} = \lambda_2 \theta_t + (1 - \lambda_2)\theta_*$ such that

$$(\theta_t - \theta_*)^T \nabla_\theta V(\theta_*)^T D(\gamma P - I)\nabla_\theta V(\theta_2^{\mathrm{mid}})(\theta_t - \theta_*) = (\theta_t - \theta_*)^T \nabla_\theta V(\theta_*)^T D(\gamma P - I)(V(\theta_t) - V(\theta_*)).$$

Using this fact, $(\theta_t - \theta_*)^T \nabla_\theta V(\theta_*)^T D(\gamma P - I)(V(\theta_t) - V(\theta_*))$ can be divided by

$$
\begin{aligned}
&(\theta_t - \theta_*)^T \nabla_\theta V(\theta_*)^T D(\gamma P - I)(V(\theta_t) - V(\theta_*)) \\
&= (\theta_t - \theta_*)^T \nabla_\theta V(\theta_*)^T D(\gamma P - I)\nabla_\theta V(\theta_*)(\theta_t - \theta_*) \\
&\quad + (\theta_t - \theta_*)^T \nabla_\theta V(\theta_*)^T D(\gamma P - I)(\nabla_\theta V(\theta_2^{\mathrm{mid}}) - \nabla_\theta V(\theta_*))(\theta_t - \theta_*) \\
&= \bar{g}_1(\theta_t) + \bar{g}_3(\theta_t).
\end{aligned}
$$

$\square$

**Lemma B.2.** *If the activation function is $l$-Lipschitz and $c_0$-smooth, for any $s$, $\theta_1$ and $\theta_2$, the inequalities*

$$||\nabla_\theta V(s, \theta_1) - \nabla_\theta V(s, \theta_2)|| \le c_0 m^{-0.5}||\theta_1 - \theta_2||,$$

$$||\nabla_\theta V(s, \theta_1)|| \le l$$

*hold where $c_1$ is a scalar that is independent of $m$ and $\theta$.*

*Proof.* Using the definition of neural network in Section 2.4, one hidden layer neural network would be simplified as

$$V(s, \theta) = \frac{1}{\sqrt{m}} \sum_{r=1}^{m} b_r \sigma(\theta^{r\,T} s).$$

It is easy to see that

$$\nabla_\theta V(s, \theta) = \frac{1}{\sqrt{m}}[b_1 \sigma'(\theta^{1\,T} s)s^T, \cdots, b_m \sigma'(\theta^{m\,T} s)s^T]^T.$$

Suppose the activation function $\sigma$ is $c_0$-smooth. That is, for any $x$ and $y$,

$$||\sigma'(x) - \sigma'(y)|| \le c_0||x - y||.$$

This means

$$
\begin{aligned}
||\nabla_\theta V(s, \theta_1) - \nabla_\theta V(s, \theta_2)||^2 &= \frac{1}{m}||[b_1(\sigma'(\theta_1^{1\,T} s) - \sigma'(\theta_2^{1\,T} s))s^T, \cdots, b_m(\sigma'(\theta_1^{m\,T} s) - \sigma'(\theta_2^{m\,T} s))s^T]||^2 \\
&\le \frac{1}{m} \sum_r ||s||^2 (\sigma'(\theta_1^{r\,T} s) - \sigma'(\theta_2^{r\,T} s))^2 \\
&\le \frac{1}{m} \sum_r ||s||^2 c_0^2 (\theta_1^{r\,T} s - \theta_2^{r\,T} s)^2 \\
&\le \frac{1}{m} \sum_r ||s||^4 c_0^2 ||\theta_1^r - \theta_2^r||^2 \\
&= \frac{c_0^2}{m}||\theta_1 - \theta_2||^2
\end{aligned}
$$

which proves the first part of the lemma. Now let us move on to the second part of the lemma. By Lipschitzness,

$$
\begin{aligned}
||\nabla V(s, \theta_1)||^2 &= \frac{1}{m} ||[b_1 \sigma'(\theta_1^{1T} s) s^T, \cdots, b_m \sigma'(\theta_1^{mT} s) s^T]||^2 \\
&\leq \frac{1}{m} \sum_r ||s||^2 (\sigma'(\theta_1^{rT} s))^2 \\
&\leq \frac{1}{m} \sum_r l^2 \\
&= l^2.
\end{aligned}
$$

Now we finish the second part of the lemma. □

**Lemma B.3.** *The following inequality holds:*

$$
\frac{\mathcal{N}(\nabla_\theta V(\theta_*)(\theta - \theta_*))}{||\theta_t - \theta_*||^2} \geq (1 - \gamma)(\sigma_{\min}^{2,D})^2.
$$

*Proof.* To begin with, the definition of $\mathcal{N}$ is

$$
\begin{aligned}
\mathcal{N}(\nabla_\theta V(\theta_*)(\theta - \theta_*)) &= (1 - \gamma)||\nabla_\theta V(\theta_*)(\theta - \theta_*)||_D^2 + \gamma ||\nabla_\theta V(\theta_*)(\theta - \theta_*)||_{\text{Dir}}^2 \\
&\geq (1 - \gamma)||\nabla_\theta V(\theta_*)(\theta - \theta_*)||_D^2.
\end{aligned}
$$

Using the definition of $\sigma_{\min}^{2,D}$,

$$
\begin{aligned}
\frac{\mathcal{N}(\nabla_\theta V(\theta_*)(\theta - \theta_*))}{||\theta - \theta_*||^2} &\geq (1 - \gamma)\frac{||\nabla_\theta V(\theta_*)(\theta - \theta_*)||_D^2}{||\theta - \theta_*||^2} \\
&\geq (1 - \gamma)(\sigma_{\min}^{2,D})^2,
\end{aligned}
$$

which establishes the result. □

**Lemma B.4.** *If a non negative sequence $\{X_t\}$ satisfies*

$$
X_{t+1} \leq (1 - \frac{c}{t+1})X_t + \frac{b}{(t+1)^2},
$$

*for some $b, c > 0$, then the sequence $\{X_t\}$ converges to 0.*

*Proof.* Recursively applying the relation between $X_{t+1}$ and $X_t$, we can derive the following:

$$
X_t \leq \frac{b}{t^2} + \sum_{i=1}^{t-1} \frac{b}{i^2} \prod_{j=i+1}^{t} (1 - \frac{c}{j}) + \prod_{j=1}^{t} (1 - \frac{c}{j}) X_0. \tag{21}
$$

The first term in Eq.(21) definitely goes to 0 as $t$ goes to infinity. Now let us consider the term $\prod_{j=i+1}^{t}(1 - \frac{c}{j})$. A logarithm usually helps us to convert it into a sum, so we perform the following manipulations:

$$
\sum_{j=i+1}^{t} \ln(1 - \frac{c}{j}) \leq - \sum_{j=i+1}^{t} \frac{c}{j} \leq c(\ln(i+1) - \ln(t+1)),
$$

where the first inequality uses the fact $\ln(1 + x) \leq x$ and the second inequality uses $\sum_{j=i+1}^{t} \frac{1}{j} \geq \ln(t+1) - \ln(i+1)$. So,

$$
\prod_{j=i+1}^{t} (1 - \frac{c}{j}) = e^{\sum_{j=i+1}^{t} \ln(1 - \frac{c}{j})} \leq e^{c \ln \frac{i+1}{t+1}} = \frac{(i+1)^c}{(t+1)^c}.
$$

By setting $i = 0$, this means the third term in Eq.(21) goes to $0$ as $t$ goes to infinity. Now consider $\sum_{i=1}^{t-1} \frac{b}{i^2} \prod_{j=i+1}^{t}(1 - \frac{c}{j})$, we obtain

$$\sum_{i=1}^{t-1} \frac{b}{i^2} \prod_{j=i+1}^{t} (1 - \frac{c}{j}) \leq \frac{b}{(t+1)^c} \sum_{i=1}^{t-1} \frac{(i+1)^c}{i^2}$$

$$\leq \frac{4b}{(t+1)^c} \sum_{i=1}^{t-1} (i+1)^{c-2}$$

$$\leq \frac{4b}{(t+1)^c} + \frac{4b}{(t+1)^c} \int_1^t (i+1)^{c-2} di,$$

where the second inequality uses the fact $\frac{(i+1)^2}{i^2} \leq 4$ for all positive integer $i$, and the third inequality combine the two facts $\sum_{i=2}^{t-1}(i+1)^{c-2} \leq \int_2^t (i+1)^{c-2} di \leq \int_1^t (i+1)^{c-2} di$ given $c \geq 2$ and $\sum_{i=2}^{t-1}(i+1)^{c-2} \leq \int_1^{t-1}(i+1)^{c-2} di \leq \int_1^t (i+1)^{c-2} di$ given $c \leq 2$. If $c \neq 1$, then

$$\sum_{i=1}^{t-1} \frac{b}{i^2} \prod_{j=i+1}^{t} (1 - \frac{c}{j}) \leq \frac{4b}{(t+1)^c} + \frac{4b}{(t+1)^c} \cdot \frac{(t+1)^{c-1} - 2^{c-1}}{c-1}.$$

If $c = 1$, then

$$\sum_{i=1}^{t-1} \frac{b}{i^2} \prod_{j=i+1}^{t} (1 - \frac{c}{j}) \leq \frac{b}{t}(1 + \ln(t+1) - \ln 2).$$

Under both cases, we can easily argue that the second term in Eq. (21) goes to $0$.

Now, we have proved that all three terms in the right hand side of Eq.(21) go to $0$ as $t$ goes to infinity. This directly implies $\{X_t\}$ converges to $0$. $\qquad \square$

**Lemma B.5.** *(Optional Stopping Theorem) Suppose $\{X_t\}$ is a super martingale and $T$ is a stopping time. If there exists a constant $c$ such that $|X_{\tau \wedge T}| \leq c$ holds for all $\tau$, then we have*

$$\mathbb{E}[X_\tau] \leq \mathbb{E}[X_0].$$

This is also called Doob's Optional Stopping Theorem. See Theorem 10.10 of Williams (1991).

### B.3  PROOF OF LEMMA B.1

Consider Non Projected Neural TD Learning,

$$\|\theta_{t+1} - \theta_*\|^2 = \|\theta_t - \theta_* + \alpha_t g(\theta_t)\|^2$$
$$= \|\theta_t - \theta_*\|^2 + 2\alpha_t(\theta_t - \theta_*)^T g(\theta_t) + \alpha_t^2 \|g(\theta_t)\|^2.$$

Given $\theta_t$, the only randomness is from $s_t$. Taking expectation on both side and using the fact that $\mathbb{E}[g(\theta_t)] = \bar{g}(\theta_t)$ (since we assume $s_t$ are sampled i.i.d. from $\mu$), we obtain

$$\mathbb{E}\left[\|\theta_{t+1} - \theta_*\|^2 | \theta_t\right] = \|\theta_t - \theta_*\|^2 + 2\alpha_t(\theta_t - \theta_*)^T \bar{g}(\theta_t) + \alpha_t^2 \mathbb{E}\left[\|g(\theta_t)\|^2 | \theta_t\right]. \tag{22}$$

First, we consider $2\alpha_t(\theta_t - \theta_*)^T \bar{g}(\theta_t)$. Lemma B.1 allows us to divide it into several parts and thus we can bound them respectively:

$$2\alpha_t(\theta_t - \theta_*)^T \bar{g}(\theta_t) = 2\alpha_t(\bar{g}_1(\theta_t) + \bar{g}_2(\theta_t) + \bar{g}_3(\theta_t)).$$

To bound $\bar{g}_1(\theta_t)$,

$$\bar{g}_1(\theta_t) = (\theta_t - \theta_*)^T \nabla_\theta V(\theta_*)^T D(\gamma P - I) \nabla_\theta V(\theta_*)(\theta_t - \theta_*)$$
$$= -\mathcal{N}(\nabla_\theta V(\theta_*)(\theta_t - \theta_*))$$
$$\leq -(1 - \gamma)(\sigma_{\min}^{2,D})^2,$$

where the first equality is because the definition of $\bar{g}_1(\theta_t)$ in Eq.(18), the second equality is by setting $f = \nabla_\theta V(\theta_*)(\theta_t - \theta_*)$ in Lemma A.1, and the inequality is by Lemma B.3.

To bound $\bar{g}_2(\theta_t)$,

$$
\begin{aligned}
\bar{g}_2(\theta_t) =& (\theta_t - \theta_*)^T (\nabla_\theta V(\theta_t) - \nabla_\theta V(\theta_*))^T D(\gamma P - I)(V(\theta_t) - V(\theta_*)) \\
\leq & \left| \gamma(\theta_t - \theta_*)^T (\nabla_\theta V(\theta_t) - \nabla_\theta V(\theta_*))^T DP(V(\theta_t) - V(\theta_*)) \right| \\
& + \left| (\theta_t - \theta_*)^T (\nabla_\theta V(\theta_t) - \nabla_\theta V(\theta_*))^T D(V(\theta_t) - V(\theta_*)) \right| \\
\leq & (1 + \gamma)c_0 l m^{-0.5} ||\theta_t - \theta_*||^3,
\end{aligned}
$$

where the equality is because the definition of $\bar{g}_2(\theta_t)$ in Eq.(19), the first inequality is by triangle inequality which says $|x + y| \leq |x| + |y|$, and the second inequality is by setting $x$ to be $(\nabla_\theta V(\theta_t) - \nabla_\theta V(\theta_*))(\theta_t - \theta_*)$, $y$ to be $V(\theta_t) - V(\hat{\theta}_*)$ in Lemma A.5 with each entry of $x, y$ bounded by $c_0 m^{-0.5}||\theta_t - \theta_*||^2, l||\theta_t - \theta_*||$ respectively using Lemma B.2.

To bound $\bar{g}_3(\theta_t)$,

$$
\begin{aligned}
\bar{g}_3(\theta_t) =& (\theta_t - \theta_*)^T \nabla_\theta V(\theta_*)^T D(\gamma P - I)(\nabla_\theta V(\theta_2^{\mathrm{mid}}) - \nabla_\theta V(\theta_*))(\theta_t - \theta_*) \\
\leq & \left| \gamma(\theta_t - \theta_*)^T \nabla_\theta V(\theta_*)^T DP(\nabla_\theta V(\theta_2^{\mathrm{mid}}) - \nabla_\theta V(\theta_*))(\theta_t - \theta_*) \right| \\
& + \left| (\theta_t - \theta_*)^T \nabla_\theta V(\theta_*)^T D(\nabla_\theta V(\theta_2^{\mathrm{mid}}) - \nabla_\theta V(\theta_*))(\theta_t - \theta_*) \right| \\
\leq & (1 + \gamma)c_0 l m^{-0.5} ||\theta_t - \theta_*||^3,
\end{aligned}
$$

where the equality is because of the definition of $\bar{g}_3(\theta_t)$ in Eq.(20), the first inequality is because of the triangle inequality, and the second inequality is by setting $x$ to be $(\nabla_\theta V(\theta_2^{\mathrm{mid}}) - \nabla_\theta V(\theta_*))(\theta_t - \theta_*)$, $y$ to be $\nabla_\theta V(\theta_*)(\theta_t - \theta_*)$ in Lemma A.5 with each entry of $x, y$ bounded by $c_0 m^{-0.5}||\theta_t - \theta_*||^2$, $l||\theta_t - \theta_*||$ respectively using Lemma B.2.

Combine the above facts and we now have the bound for the second term in the right hand side of Eq.(22),

$$
2\alpha_t(\theta_t - \theta_*)^T \bar{g}(\theta_t) \leq -2\alpha_t(1 - \gamma)(\sigma_{\min}^{2,D})^2 ||\theta_t - \theta_*||^2 + 4\alpha_t(1 + \gamma)c_0 l m^{-0.5}||\theta_t - \theta_*||^3.
$$

Second, we consider $\mathbb{E}[\alpha_t^2 ||g(\theta_t)||^2 | \theta_t]$ in Eq.(22). For simplicity, define $f(s) = V(s, \theta_*) - V(s, \theta_t)$. Since we are using one hidden layer neural network to approximate $V(s, \theta)$,

$$
\begin{aligned}
|f(s)|^2 =& |V(s, \theta_*) - V(s, \theta_t)|^2 \\
=& \frac{1}{m} \left| \sum_r b_r(\sigma(\theta_t^r s) - \sigma(\theta_*^r s)) \right|^2 \\
\leq & \sum_r b_r^2 ||\sigma(\theta_t^r s) - \sigma(\theta_*^r s)||^2 \\
\leq & \sum_r l^2 ||\theta_t^r s - \theta_*^r s||^2 \\
\leq & \sum_r l^2 ||\theta_t^r - \theta_*^r||^2 \\
=& l^2 ||\theta_t - \theta_*||^2.
\end{aligned}
$$

Recall that $V(s, \theta_*)$ satisfies Eq.(1), which is

$$
V(s, \theta_*) = r(s) + \gamma \sum_{s''} P(s''|s)V(s'', \theta_*),
$$

and $g(\theta_t)$ is defined to be $g(\theta_t) = \nabla_\theta V(s, \theta_t) [r(s) + \gamma V(s', \theta_t) - V(s, \theta_t)]$. This latter immediately implies that $g(\theta_t)$ is actually a random variable and implicitly relies on the state $s$. Using these facts, we obtain

$$
\begin{aligned}
g(\theta_t) =& \nabla_\theta V(s, \theta_t) [r(s) + \gamma V(s', \theta_t) - V(s, \theta_t)] \\
=& \nabla_\theta V(s, \theta_t) \left[ f(s) - \gamma \sum_{s''} P(s''|s)f(s') + \gamma \sum_{s''} P(s''|s)(V(s', \theta_*) - V(s'', \theta_*)) \right].
\end{aligned}
$$

By Eq.(2) we can obtain the following quick result:

$$|V(s', \theta_*) - V(s'', \theta_*)| \leq \frac{2r_{\max}}{1 - \gamma}.$$

And this leads to the following:

$$
\begin{aligned}
||g(\theta_t)||^2 &= \left\| \nabla_\theta V(s, \theta_t) \left[ f(s) - \gamma \sum_{s''} P(s''|s) f(s') + \gamma \sum_{s''} P(s''|s)(V(s', \theta_*) - V(s'', \theta_*)) \right] \right\|^2 \\
&\leq 3||\nabla_\theta V(s, \theta_t)||^2 \left[ |f(s)|^2 + \left| \gamma \sum_{s''} P(s''|s) f(s') \right|^2 + \left| \gamma \sum_{s''} P(s''|s)(V(s', \theta_*) - V(s'', \theta_*)) \right|^2 \right] \\
&\leq 3||\nabla_\theta V(s, \theta_t)||^2 \left[ |f(s)|^2 + \gamma^2 \sum_{s''} P(s''|s)|f(s')|^2 + \gamma^2 \sum_{s''} P(s''|s)|V(s', \theta_*) - V(s'', \theta_*)|^2 \right] \\
&\leq 3(1 + \gamma^2) l^4 ||\theta_t - \theta_*||^2 + \frac{12\gamma^2 l^2 r_{\max}^2}{(1 - \gamma)^2},
\end{aligned}
\tag{23}
$$

where the first inequality uses the fact that $(a + b + c)^2 \leq 3(a^2 + b^2 + c^2)$, the second inequality is by Jensen's inequality, and the third inequality simply uses Lemma B.2 to bound $||\nabla_\theta V(s, \theta_t)||$. Further,

$$\alpha_t^2 \mathbb{E}\left[||g(\theta_t)||^2 | \theta_t\right] \leq 3\alpha_t^2(1 + \gamma^2) l^4 ||\theta_t - \theta_*||^2 + \frac{12\alpha_t^2 \gamma^2 l^2 r_{\max}^2}{(1 - \gamma)^2}.$$

Now let us go back to Eq.(22), where the second and third terms on the right hand side can be bounded by

$$
\begin{aligned}
&2\alpha_t(\theta_t - \theta_*)^T g(\theta_t) + \alpha_t^2 \mathbb{E}\left[||g(\theta_t)||^2 | \theta_t\right] \\
&\leq -2\alpha_t(1 - \gamma)(\sigma_{\min}^{2,D})^2 ||\theta_t - \theta_*||^2 + 4\alpha_t(1 + \gamma) c_0 l m^{-0.5} ||\theta_t - \theta_*||^3 \\
&\quad + 3\alpha_t^2(1 + \gamma^2) l^4 ||\theta_t - \theta_*||^2 + \frac{12\alpha_t^2 \gamma^2 l^2 r_{\max}^2}{(1 - \gamma)^2} \\
&= \left( -2\alpha_t(1 - \gamma)(\sigma_{\min}^{2,D})^2 + 3\alpha_t^2(1 + \gamma^2) l^4 \right) ||\theta_t - \theta_*||^2 \\
&\quad + 4\alpha_t(1 + \gamma) c_0 l m^{-0.5} ||\theta_t - \theta_*||^3 + \frac{12\alpha_t^2 \gamma^2 l^2 r_{\max}^2}{(1 - \gamma)^2}.
\end{aligned}
$$

This means

$$
\begin{aligned}
&\mathbb{E}\left[||\theta_{t+1} - \theta_*||^2 | \theta_t\right] \\
&\leq \left( 1 - 2\alpha_t(1 - \gamma)(\sigma_{\min}^{2,D})^2 + 3\alpha_t^2(1 + \gamma^2) l^4 \right) ||\theta_t - \theta_*||^2 \\
&\quad + 4\alpha_t(1 + \gamma) c_0 l m^{-0.5} ||\theta_t - \theta_*||^3 + \frac{12\alpha_t^2 \gamma^2 l^2 r_{\max}^2}{(1 - \gamma)^2}.
\end{aligned}
\tag{24}
$$

For simplicity, we define the following notations:

$$X_t = ||\theta_t - \theta_*||,$$
$$C = \frac{12\gamma^2 l^2 r_{\max}^2}{\lambda^2(1-\gamma)^2},$$
$$\alpha_t = \frac{1}{\lambda(t+1)},$$
$$a_t = 2\alpha_t(1-\gamma)(\sigma_{\min}^{2,D})^2 - 3\alpha_t^2(1+\gamma^2)l^4,$$
$$b_t = 4\alpha_t(1+\gamma)c_0 l m^{-0.5},$$
$$c_t = \frac{12\alpha_t^2\gamma^2 l^2 r_{\max}^2}{(1-\gamma)^2} = \frac{C}{(t+1)^2},$$
$$d_t = \frac{C}{t} \quad \text{(while } d_0 \text{ is defined to be } d_0 = 2).$$

We can rewrite Eq.(24) as

$$\mathbb{E}\left[X_{t+1}^2|X_t\right] \leq (1 - a_t + b_t X_t)X_t^2 + c_t,$$

while the condition $\mathbb{E}\left[||\theta_0 - \theta_*||^2\right] < \frac{\left(2(1-\gamma)(\sigma_{\min}^{2,D})^2\lambda - 3l^4(1+\gamma^2)\right)^2}{64(1+\gamma)^2 c_0^2 l^2 \lambda^2}m\delta - 2$ is just

$$\mathbb{E}[X_0^2] \leq \frac{a_0^2}{4b_0^2}\delta - d_0.$$

Under such notations, an important fact is that

$$c_t \leq d_t - d_{t+1}.$$

This can be showed easily since $\frac{1}{(t+1)^2} \leq \frac{1}{t} - \frac{1}{t+1}$.

If we define the stopping time $T = \inf_t\{X_t \geq \frac{a_0}{2b_0}\}$ and the sequence $\{Y_t\}$ to be

$$Y_t = \begin{cases} X_t^2 + d_t & \text{if } t \leq T \\ Y_{t-1} & \text{if } t > T \end{cases}$$

Now we first claim that the sequence $\{Y_t\}$ is a super martingale. We show this as follows:

Suppose $T = 0$, and we find that $Y_t = Y_0, \forall t$. In this trivial case, obviously, $\{Y_t\}$ is a super martingale. Now we assume $T > 0$.

When $t = 0$, we have

$$\mathbb{E}[X_1^2|X_0] \leq (1 - \frac{1}{2}a_0)X_0^2 + c_0 \leq X_0^2 + c_0 \leq X_0^2 + d_0 - d_1.$$

which means $\mathbb{E}[Y_1|Y_0] \leq Y_0$.

For $t \leq T$, by the definition of $T$ we know that $X_t \leq \frac{a_0}{2b_0}$, which implies that $X_t \leq \frac{a_t}{2b_t}$ (this is because by the definition of $a_t$ and $b_t$, $h(t) = \frac{a_t}{2b_t}$ is increasing with $t$). Hence,

$$\mathbb{E}[X_{t+1}^2|X_t] \leq (1 - \frac{1}{2}a_t)X_t^2 + c_t \leq X_t^2 + c_t \leq X_t^2 + d_t - d_{t+1}.$$

which means $\mathbb{E}[Y_{t+1}|Y_t] \leq Y_t$.

For $t > T$, by the definition of $\{Y_t\}$ we know that $Y_t = Y_T$.

Hence, combine all the above facts and we conclude the sequence $\{Y_t\}$ is a super martingale.

Next, we claim that $X_t < \frac{a_0}{2b_0}, \forall t$ holds with probability at least $1 - \delta$. This can be shown as follows:

Let $A$ be the event $\left\{\sup_t X_t < \frac{a_0}{2b_0}\right\} = \left\{X_0 < \frac{a_0}{2b_0}, X_1 < \frac{a_0}{2b_0}, \cdots\right\}$. To compute $P(A)$, notice that

$$P\left(X_0 < \frac{a_0}{2b_0}, X_1 < \frac{a_0}{2b_0}, \cdots\right) = P\left(Y_0 < \frac{a_0^2}{4b_0^2} + d_0, Y_1 < \frac{a_0^2}{4b_0^2} + d_1, \cdots\right).$$

We can easily check that $T$ is also a stopping time for $\{Y_t\}$. In order to use Lemma B.5, we need to check the conditions that $|Y_T| \leq c$ for some constant $c$. We split it into two cases.

First, if $\tau < T$, then $|Y_{\tau \wedge T}| = |Y_\tau|$. Because of the definition of $T$ we know that

$$|Y_\tau| \leq \frac{a_0^2}{4b_0^2} + d_\tau \leq \frac{a_0^2}{4b_0^2} + 2,$$

where we use the fact that $\{d_t\}$ is a decreasing sequence and $d_0 = 2$.

Second, if $\tau \geq T$, then $|Y_{\tau \wedge T}| = |Y_T|$. Recall that update of the algorithm is

$$\theta_{t+1} = \theta_t + \alpha_t g(\theta_t).$$

which implies

$$||\theta_{t+1} - \theta_*|| \leq ||\theta_t - \theta_*|| + \alpha_t ||g(\theta_t)||.$$

Now we let $t = T - 1$ and use $X_t$ notations. The above fact can be rewritten as

$$|X_T| \leq |X_{T-1}| + \alpha_{T-1} ||g(\theta_{T-1})||.$$

Because of the definition of $T$, we know that $|X_{T-1}| \leq \frac{a_0}{2b_0}$. Moreover, Eq.(23) give us a bound for $||g(\theta_{T-1})||$, which is $||g(\theta_{T-1})|| \leq \sqrt{3(1+\gamma^2)l^4 X_{T-1}^2 + \frac{12\gamma^2 l^2 r_{\max}^2}{(1-\gamma)^2}}$. Finally, $\alpha_{T-1}$ is set to be $\frac{1}{\lambda(t+1)}$ so it is obviously bounded. All these facts lead to the result that $|Y_T|$ is bounded.

Combining the above two cases, we are now eligible to use Lemma B.5. Hence, we obtain

$$\mathbb{E}[Y_{\tau \wedge T}] \leq \mathbb{E}[Y_0].$$

On the other hand, $\mathbb{E}[Y_{\tau \wedge T}]$ can be expanded as

$$\mathbb{E}[Y_{\tau \wedge T}] = \mathbb{E}[Y_{\tau \wedge T}|T \leq \tau]P(T \leq \tau) + \mathbb{E}[Y_{\tau \wedge T}|T > \tau]P(T > \tau)$$
$$= \mathbb{E}[Y_T]P(T \leq \tau) + \mathbb{E}[Y_\tau]P(T > \tau).$$

Combining these two facts,

$$\mathbb{E}[Y_0] \geq \mathbb{E}[Y_T]P(T \leq \tau)$$
$$\geq \left(\frac{a_0^2}{4b_0^2} + d_T\right) P(T \leq \tau).$$

where the second inequality using the fact that $Y_t \geq \frac{a_0^2}{4b_0^2} + d_T$. Hence,

$$P(T \leq \tau) \leq \frac{\mathbb{E}[Y_0]}{\frac{a_0^2}{4b_0^2} + d_T} = \frac{\mathbb{E}[X_0^2] + d_0}{\frac{a_0^2}{4b_0^2} + d_T} \leq \frac{\mathbb{E}[X_0^2] + d_0}{\frac{a_0^2}{4b_0^2}} \leq \delta.$$

where the equality is because $\mathbb{E}[Y_0] = \mathbb{E}[X_0^2] + d_0$, the second inequality is because $d_t$ is nonnegtive, and the third inequality is because of the condition $\mathbb{E}[X_0^2] \leq \frac{a_0^2}{4b_0^2}\delta - d_0$. Next, it is easy to see

$$P\left(Y_0 < \frac{a_0^2}{4b_0^2} + d_0, Y_1 < \frac{a_0^2}{4b_0^2} + d_1, \cdots, Y_\tau < \frac{a_0^2}{4b_0^2} + d_\tau\right) = P(T > \tau)$$
$$= 1 - P(T \leq \tau)$$
$$\geq 1 - \delta.$$

This result holds for all $\tau$, so we can let $\tau \to \infty$ and obtain

$$P(A) \geq 1 - \delta.$$

Actually, using $h(t) = \frac{a_t}{2b_t}$ is monotonically increase with $t$ again, event $A$ implies $X_t \leq \frac{a_t}{2b_t}, \forall t$. With this fact,

$$\mathbb{E}\left[X_{t+1}^2 | X_t, A\right] \leq (1 - a_t + b_t X_t)X_t^2 + c_t \leq (1 - \frac{1}{2}a_t)X_t^2 + c_t, \forall t.$$

Plugging in $a_t = 2\alpha_t(1-\gamma)(\sigma_{\min}^{2,D})^2 - 3\alpha_t^2(1+\gamma^2)l^4$, $c_t = \frac{12\alpha_t^2\gamma^2l^2r_{\max}^2}{(1-\gamma)^2}$ and $\alpha_t = \frac{1}{\lambda(t+1)}$, we can derive

$$\mathbb{E}\left[X_{t+1}^2|X_t, A\right] \leq \left[1 - \frac{(1-\gamma)(\sigma_{\min}^{2,D})^2}{\lambda(t+1)} + \frac{3(1+\gamma^2)l^4}{2\lambda^2(t+1)^2}\right]X_t^2 + \frac{12\gamma^2l^2r_{\max}^2}{(1-\gamma)^2\lambda^2(t+1)^2}.$$

Given that $X_t \leq \frac{a_0}{2b_0} \leq \frac{(1-\gamma)m^{0.5}(\sigma_{\min}^{2,D})^2}{2(1+\gamma)c_0l}$. Hence, we conclude

$$\mathbb{E}\left[X_{t+1}^2|X_t, A\right] \leq \left[1 - \frac{(1-\gamma)(\sigma_{\min}^{2,D})^2}{\lambda(t+1)}\right]X_t^2 + \frac{3(1-\gamma)^2(1+\gamma^2)l^2(\sigma_{\min}^{2,D})^2m}{8\lambda^2(1+\gamma)^2c_0^2(t+1)^2} + \frac{12\gamma^2l^2r_{\max}^2}{(1-\gamma)^2\lambda^2(t+1)^2}.$$

Take expectation on both sides and we derive

$$\mathbb{E}\left[X_{t+1}^2|A\right] \leq \left[1 - \frac{(1-\gamma)(\sigma_{\min}^{2,D})^2}{\lambda(t+1)}\right]\mathbb{E}[X_t^2|A] + \frac{3(1-\gamma)^2(1+\gamma^2)l^2(\sigma_{\min}^{2,D})^2m}{8\lambda^2(1+\gamma)^2c_0^2(t+1)^2} + \frac{12\gamma^2l^2r_{\max}^2}{(1-\gamma)^2\lambda^2(t+1)^2}.$$

By Lemma B.4, we conclude that the sequence $\left\{\mathbb{E}\left[X_t^2|A\right]\right\}$ converges to 0.

