# OpenReview forum: "On the Performance of Temporal Difference Learning With Neural Networks"
_ICLR.cc/2023/Conference — ICLR 2023 poster_

### Official Review · Reviewer_ZfL8 · 2022-10-18

**Confidence:** 2
**Correctness:** 4
**Technical Novelty And Significance:** 3
**Empirical Novelty And Significance:** 2
**Recommendation:** 8

**Clarity, Quality, Novelty And Reproducibility:**

Overall the paper reads very well. However, I did not check the proof of the theorem in detail.

**Strength And Weaknesses:**

Strengths:
+ Given the wide usage of neural TD this paper provides a great contribution for the community
+ The paper is well-written overall while being a theoretical paper

Weaknesses
- More tips around the applicability of the theorems to practitioners would have been great. For example ReLU, a widely used function, does not satisfy the assumption of the theorem.
- I could not find the constant value of the omega for the results shown.


Minor errors:
Page 2: pi(s,a) -> \pi (s,a)
Page 4: \sigma is and activation -> \sigma is an activation

**Summary Of The Paper:**

Authors derived a new bound for Neural TD projecting for a fixed policy using arbitrary set of layers within the ball with radius \omega around a starting parameter \theta_0. They showed that the approximation error is bounded by O(\epsilon + 1 / \sqrt(m)) where m is the width of the neural network and \epsilon is the approximation quality of the best neural network within that ball. The main advantage of the new theorem is that it relaxed four conditions of the previous proofs: 1) small projection radius, 2) linearization around the initial condition, 3) restriction on policies, and 4) limited to a single layer network. Authors provided insight on the effect of m by running experiments for policy evaluation across three toy domains with both a random and well-trained policies.

**Summary Of The Review:**

Overall a great and well-written paper with a major theoretical step.

---

> ### Author Response · Authors · 2022-11-14
> **Reply to reviewer ZfL8**
>
> > Given the wide usage of neural TD this paper provides a great contribution for the community
> The paper is well-written overall while being a theoretical paper.
>
> Thank you for the positive assessment of our work! This paper is the culmination of over two years of work on our part trying to understand and clarify the literature of neural TD.
>
> > More tips around the applicability of the theorems to practitioners would have been great. For example ReLU, a widely used function, does not satisfy the assumption of the theorem.
>
> Indeed, we acknowledge this is a shortcoming of our results, and we have included a discussion of this in the paper. From a technical point of view, our arguments require the underlying function to be smooth, which is why we make this assumption. From a practical point of view, if one wants to use this with ReLU activation, a small modification that one can make is use a smooth approximation such as a GeLU instead (\url{https://alaaalatif.github.io/gelu_imgs/gelu_viz-1.png}), which is unlikely to make much of a difference to the result.
>
>  It is possible to have a result with ReLU activations, but then one needs to assume that the state is approximately random under the stationary distribution so that the update is ``smooth on the average'' (randomness means states are unlikely to fall near the point where the derivative of the ReLU changes, so that randomness acts as a smoother). This is what was done in the previous work Cai et al. (2019). Since we do not want to assume randomness of the state, we have to stick with smooth activations.
>
> > I could not find the constant value of the $\omega$ for the results shown.
>
> The constant $\omega$ can be anything! But then it appears in various bounds we obtain, so any choice of $\omega$ allows it to be incorporated into the $O(\cdot)$ notation. What is not allowed -- in the sense that it would require modifying the arguments -- is to choose $\omega = \sqrt{m}$ or some such.

---

### Official Review · Reviewer_bfdn · 2022-10-23

**Confidence:** 4
**Clarity, Quality, Novelty And Reproducibility:** The writing of the paper is clear and…
**Correctness:** 4
**Technical Novelty And Significance:** 4
**Empirical Novelty And Significance:** Not applicable
**Recommendation:** 6

**Strength And Weaknesses:**

Strength: the paper uses new techniques to analyze the convergence of neural TD. In particular, it utilize a combination of D-norm and Dirichlet norm to characterize the distance to the true value function. The reviewer believes that the analysis brings insight to the reinforcement learning community.

Weakness: The theoretical result seems a weak increment. It is not clear why changing the projection radius to constant can help the TD learning in practice. In fact, since the essential models considered are still linear models implied by the neural tangent kernel (NTK), the generalization error will be the same as the settings of previous neural TD works. From the computation cost perspective, it will be more interesting if the authors can show that the projection step can be totally discarded (just as most NTK works on supervised works do).

**Summary Of The Paper:**

This paper studies the convergence of the temporal-difference (TD) algorithm with overparametrized neural networks. Compared with previous works, it weakens the projection step radius to a constant level by using a different analyzing techniques. The paper also provides simulation results to support the theories.

**Summary Of The Review:**

The paper uses novel techniques to modify the previous analysis of the global convergence of neural TD. The reviewer believes that the result, although not quite strong, is helpful to the understanding of such a problem.

---

> ### Author Response · Authors · 2022-11-14
> **Reply to reviewer bfdn: part I**
>
> > Strength: the paper uses new techniques to analyze the convergence of neural TD. In particular, it utilize a combination of D-norm and Dirichlet norm to characterize the distance to the true value function. The reviewer believes that the analysis brings insight to the reinforcement learning community.
>
> Thank you for the positive assessment of our work!
>
> > Weakness: The theoretical result seems a weak increment. It is not clear why changing the projection radius to constant can help the TD learning in practice. In fact, since the essential models considered are still linear models implied by the neural tangent kernel (NTK), the generalization error will be the same as the settings of previous neural TD works).
>
> This is an interesting remark, and we thank the reviewer for bringing this is up -- this gives us a chance to discuss the placement of this paper within NTK theory.
>
> As a first response, though, note that *our simulations show a strong improvement from choosing the projection radius to be a constant!* Indeed, in Section 4, all the solid lines (constant projection radius) outperform the dashed lines ($ \sim m^{-1/2} $ projection radius) of the same color.
>
> Now typical NTK analysis proceeds by arguing that, when the network is wide, it stays close to the initial condition during trainining, and is consequently approximately linear. The cleanest statement of this is in https://proceedings.neurips.cc/paper/2019/hash/ae614c557843b1df326cb29c57225459-Abstract.html, and previous papers on neural TD and neural Q-learning generally used that approach. But we do not see this in training. In fact, here is an additional set of graphs we generated that show the distance to the initial condition divided by the projection radius:
>
> https://anonymous.4open.science/r/images_in_paper-87D6/dist.png
>
> As can be seen, the iterates move to the *boundary* of the projection region in every case. In other words, the neural TD iterates move as far from the initial condition as they can.
>
> A partial explanation for this is that the width ($m=80$ to $m=160$) that we are simulating (with depths ranging from $3$ to $7$) is likely not large enough to be covered by the NTK-type results that ensure closeness to initial condition. As a further test of this, we plot the error measure $$||\nabla V_{\theta_t}(s_t) - \nabla V_{\theta_0} (s_t)||,$$ which measures the linearity of the network during training: for a linear map, this measure is identically zero because the gradient would be constant. The results are at
>
> https://anonymous.4open.science/r/images_in_paper-87D6/grad.png
>
> Note that we show the graph starting at iteration/$20=1$, so that the graphs do not begin at zero. We see that the nonlinearity increases during training and is not close to zero, even for the networks with a small projection radius.
>
> **To summarize:** choosing a constant projection radius strongly improves performance, results in approximations that are nonlinear and that move as far away as they can from their initial condition. These simulations (coupled with simulations that show improved performance already in the body of the paper) show the value of using a constant projection radius that does not decay with $m$ and also show why the model is not linear.
>
> It is natural to wonder how this can be: indeed, the reviewers comments are supported by a large amount of NTK theory. We believe the answer is that the latest analyses of wide neural networks proceeded by showing that bounds on the nonlinearity decay with $m$, allowing NTK-type proofs to be applicable to much lower values of the width $m$, where the network is still sufficiently nonlinear to have low generalization error. The primary paper we rely on for this kind of analysis is  https://proceedings.neurips.cc/paper/2020/hash/b7ae8fecf15b8b6c3c69eceae636d203-Abstract.html.
>
>
> Indeed, *one of the main novel things about this paper is that it is fundamentally nonlinear*. We compare our result to the best *nonlinear* neural network and achieve a decaying error in the number of iterations $T$ and the width $m$. We do this without any kind of representability assumption on the underlying policy. By contrast, previous papers on the subject made assumptions that explicitly forced the network to stay close to the initial condition. We argue that this is a conceptual improvement to the state-of-the-art and wonder if the reviewer might reconsider their characterization of this work as a ``weak increment."

---

> ### Author Response · Authors · 2022-11-14
> **Reply to reviewer bfdn: part II**
>
> > From the computation cost perspective, it will be more interesting if the authors can show that the projection step can be totally discarded (just as most NTK works on supervised works do).
>
> We have actually added this to the paper! Please see Section B of the supplementary information for an analysis of the unprojected case, though only for a single hidden layer. However, we want to push back a little here: we don't want to do what most NTK works do, which is reduce the wide-case to something that stays close to initialization.
>
> The theorem we have added to the draft  that if the distance between $\theta_0$ and $\theta^*$ is at most $C \delta \sqrt{m}$, for some explicitly written out constant $C$,  then with probability $1-\delta$, the iterates remain bounded, which we call event ${\cal A}$; we then show that $E[||\theta_t - \theta^*||^2 | {\cal A}] \rightarrow 0$.
>
> Note that a distance that scales as $\sim \sqrt{m}$ allows a constant error on every single coefficient of the neural network.
>
> We do not believe it is possible to remove the projection entirely, not without assuming $m$ is so large that we stay in the ``linear'' regime. Indeed, our simulations for values of $m=80$ to $m=160$ with depths ranging from $3$ to $7$ already show examples of divergence without projection.

---

### Official Review · Reviewer_SRvC · 2022-10-24

**Confidence:** 3
**Correctness:** 3
**Technical Novelty And Significance:** 2
**Empirical Novelty And Significance:** 2
**Recommendation:** 6

**Clarity, Quality, Novelty And Reproducibility:**

The paper is clearly written, and includes somewhat new results.
Some points need to be clarified, and they are given in the previous comments.

**Strength And Weaknesses:**

The paper provides some new results on the convergence of TD learnign with neural network.
The results seem interesting.
Weaknesses of the paper are summarized as follows:
1) The definition of semi-norm in eq (5) is not defined.
Moreover, the definition caligraphic N would not be a norm. It is a semi-norm.
Some discusssions would be needed.
2) In the definition of fully connected neural network, is there a bias term? Usually, NN has bias parameter in each layer.
It should be clarified.
3) It is not discussed why the final weight b is fixed.
4) Assumption 2.4. requires that activation function is l-Lipschitz. However, to my knowledge, most popular activation functions are not globally Lipschitz.
Some discusssions would be needed.
5) In the result of Theorem 3.1, caligraphic N is not a norm. Therefore, we may not have any useful convergence result using the semi-norm because semi-norm may be zero for nonzero input.
6) Moreover, in the left-hand side, the average cannot go inside the V since it is nonlinear. Then how can we derive a conclusion for theta_t?


**Summary Of The Paper:**

The paper provides a convergence analysis of Neural TD Learning with a projection onto a ball of fixed radius around the initial point.

**Summary Of The Review:**

The paper seems to contain interesting results.
However, some assumptions are strong to be practical, and some convergence results need to be clarified more as indicated in my previous comments.

---

> ### Author Response · Authors · 2022-11-14
> **Reply to reviewer SRvC: part I**
>
> We would like to thank the reviewer for their valuable comments, which have led us to make a number of revisions and clarifications to the paper. Our point-by-point response to the comments is given next.
>
> > The definition of semi-norm in eq (5) is not defined. Moreover, the definition calligraphic $N$ would not be a norm. It is a semi-norm. Some discussions would be needed.
>
> > In the result of Theorem 3.1, calligraphic $N$ is not a norm. Therefore, we may not have any useful convergence result using the semi-norm because semi-norm may be zero for nonzero input.
>
> Thank you for pointing this out -- *this gives us a chance to clarify that the calligraphic $N$ is a norm!* We have edited the paper to state this explicitly. More precisely, the calligraphic $N$ is actually the square of a norm. Please see the remarks in red in the paper revision immediately after the definition of $N$ on the top of page four.
>
> Let us use this opportunity to spell out what upper bounds for ${\cal N}(\cdot)$ means in comparison to the previous work. The standard analysis of TD learning provides bounds in $D$-norm, $$ ||V||_D^2 = \sum_s \mu(s) V(s)^2.$$ For example, the first theoretical analysis of TD in *[Tsitsiklis, Van Roy, Analysis of temporal-difference learning with function approximation, NeurIPS 96]* used the $D$-norm, and this has been repeated in analyses since. The core reason for this is that, in the linear approximation case, the Bellman operator is non-expansive in the $D$-norm.
>
> Now we instead use the calligraphic $N$, defined as
>
> $${\cal N}(V)= (1-\gamma)  ||V||_D^2 + \gamma ||V||_R^2,$$
>
> where we use $||V||_R^2$ for the squared Dirichlet semi-norm, since OpenReview seems to only allow single-character subscripts. But clearly,
> $$ {\cal N}(V) \geq (1-\gamma) ||V||_D^2,$$ obtained by just ignoring the second term. Therefore any upper bound on ${\cal N}(V)$ is also an upper bound on $||V||_D^2$, with the addition of a $(1-\gamma)^{-1}$ multiplicative term. Incidentally, this is why $(1-\gamma)^{-1}$ terms, which appeared in the bounds of all the previous papers, are missing from the right-hand sides of our bounds: they are actually here as well, but moved to the left-hand side as they are included in the definition of ${\cal N}(\cdot)$.
>
> To summarize, **our analysis is in a norm, and moreover our bounds may be re-arranged to give bounds on the ``standard'' norm used to bound performance of TD.** We have edited the paper to point this out -- see Sec. 2.3, and the discussion immediately after Theorem 3.1.
>
> > In the definition of fully connected neural network, is there a bias term? Usually, NN has bias parameter in each layer. It should be clarified.
>
> We have now added a paragraph to discuss this definition. Our NN definition can encompass a bias term.  Indeed, if the encoding of states into vectors is such that the last entry of $x^{(0)}$ is $1$, and if, for every single layer, the matrix $\theta^{(k)}$ has $\begin{pmatrix} 0 & \cdots & 0 & 1 \end{pmatrix}$ as the last row, then this is mathematically equivalent to having a bias term. So nothing is actually lost by using our more compact definition which does not add a bias term explicitly. We also point out that previous works  i.e., Xu \& Gu 2020) and (Liu et al. 2020) (references same as in our paper) used the same more compact definition, which is why we have also adopted it.
>
> > It is not discussed why the final weight $b$ is fixed.
>
> Likewise, we have followed the previous literature i.e. (Cai et al. 2019), (Allen-Zhu et al. 2018c), (Liu et al. 2020) on this point. The reason is that the approximation results work without training the last layer -- and since altering the last layer does not appear to be needed, it is logical not to do it.

---

> > ### Comment · Reviewer_SRvC · 2022-12-14
> > **Response to the authors' response**
> >
> > I thank the authors for the clarification. The authors addressed my previous comments well. I will increase the score to a acceptable level.

---

> ### Author Response · Authors · 2022-11-14
> **Reply to reviewer SRvC: part II**
>
>
> > Assumption 2.4. requires that activation function is l-Lipschitz. However, to my knowledge, most popular activation functions are not globally Lipschitz. Some discussions would be needed.
>
> Indeed, this is relevant point, and we have inserted a discussion into the manuscript. From a technical point of view, our arguments require the underlying function to be smooth, which is why we make this assumption (we assume this is what the reviewer meant here -- ReLU function is Lipschitz, but not smooth, and we require both). From a practical point of view, if one wants to use this with ReLU activation, a small modification that one can make is use a smooth approximation such as a GeLU instead ( https://alaaalatif.github.io/gelu_imgs/gelu_viz-1.png ), which is unlikely to make much of a difference to the result.
>
> It is possible to have a result with ReLU activations, but then one needs to assume that the state is approximately random under the stationary distribution so that the update is ``smooth on the average'' (randomness means states are unlikely to fall near the point where the derivative of the ReLU changes, so that randomness acts as a smoother). This is what was done in the previous work Cai et al. (2019). Since we do not want to assume randomness of the state, we have to stick with smooth activations.
>
>
> > Moreover, in the left-hand side, the average cannot go inside the V since it is nonlinear. Then how can we derive a conclusion for $\theta_t$?
>
> That is correct: the left hand side of the main theorem is not linear with respect to $\theta$, so as the reviewer points out, we cannot put the average inside.
> However, it is enough to get that the error corresponding to a *random iterate* on $1, \ldots, T$ goes to zero -- please see the second to last paragraph on page 6 of our paper. So this does lead to an algorithm: run projected TD for $T$ steps, then take a uniformly random iterate among those generated.
>
> We acknowledge the reviewer's point that this is less nice than a bound on the last iterate $\theta_T$.  Unfortunately, this is a common issue in stochastic optimization going beyond RL -- in many contexts it is possible to derive a bound on the error corresponding to a random iterate but quite challenging to derive a bound on the final iterate.
>
>
>
> **To summarize:** given that we have clarified the main concern raised by the reviewer -- *namely, whether we are bounding an actual norm* -- we are wondering if the reviewer might be willing to re-evaluate their score of our paper. Regardless, we would like to thank the reviewer once again for their comments which led us to improve the paper in several significant ways.

---

### Official Review · Reviewer_oRiv · 2022-11-08

**Confidence:** 4
**Correctness:** 3
**Technical Novelty And Significance:** 2
**Empirical Novelty And Significance:** Not applicable
**Recommendation:** 6

**Clarity, Quality, Novelty And Reproducibility:**

Main body of the paper is clearly written - see comments above regarding the Appendix.

**Strength And Weaknesses:**

I find the analysis and results of the paper interesting - quantifying the first order progress term using arguments from gradient splitting is a good idea. However, I feel the extension from linear -> neural network function approximation is fairly straightforward from a technical point of view. Straightforward extensions can make for good papers - my main complaint is that the authors need to do a much better job of putting their contributions in perspective of prior results. Some comments:

1) The three relevant papers are Cai et. al. (2019), Xu and Gu (2020) and Cayci et. al. (2021). The authors reference problems in that analysis - that the projection radius needs to shrink at a rate of $\mathcal{O}(m^{-1/2})$ -- but don't specifically compare the results in these papers to their results presented in Theorem 1.

My understanding is that most of these papers also characterize the quality of solutions that Neural TD converges to - for example, see section 4 in Cai et. al. (2019) - the approximation error here may be non-vanishing but there is a characterization of the limit point in terms of the solutions of the projected Bellman equation. The results in Cayci et. al. (2021) seems stronger. In addition to provide a projection free analysis, they also give guidance about scaling the network width for a given value of target error. It seems that in their results, $\epsilon$ can be arbitrary large and there is now guidance on how to select $\omega$.

I request the authors to provide a more detailed, clear and transparent comparison to past work - highlighting the strengths and weaknesses of their approach as well as results vis-a-vis past work. That would be immensely useful. Maybe an empirical comparison to projection free and max-norm scaling methods of Cayci et. al. (2021) is also useful.

2) While the main paper is well written, I think the Appendix requires some work in clear writing. Instead of saying 'By Lemma A.5 and eq(13)' can you kindly write out each expression and bound term by term? Can the equations be properly numbered to show how '$g(\theta_t)$ can be rewritten as' .. Similar comments go for the proof of non i.i.d case. I assume most inequalities to be correct (since the resulting bounds make sense), I don't think these have been properly justified. I think it is really bad to put the onus on reviewers to parse badly written proofs - the burden must be on the authors to write in a clear and easy to follow manner.




**Summary Of The Paper:**

This paper provides a convergence analysis of projected Neural TD - Temporal Difference learning using a neural network for function approximation, and a projection step onto a norm ball of fixed radius around the initial iterate. The authors show sublinear convergence with an approximation error of $\mathcal{O}(\epsilon + \frac{1}{\sqrt{m}})$, where $m$ is the width of all hidden layers of the network. The paper mainly builds on the idea of ``gradient splitting'' which was previously used to analyze TD learning with linear function approximation [Liu and Olshevsky, 2021].

**Summary Of The Review:**

I am inclined to recommend accepting this work conditional on the authors putting their results in better perspective as compared to prior work as well as improving proofs in the Appendix.

---

> ### Author Response · Authors · 2022-11-14
> **Reply to reviewer oRiv: part I**
>
> > I find the analysis and results of the paper interesting - quantifying the first order progress term using arguments from gradient splitting is a good idea. However, I feel the extension from linear -> neural network function approximation is fairly straightforward from a technical point of view. Straightforward extensions can make for good papers - my main complaint is that the authors need to do a much better job of putting their contributions in perspective of prior results. Some comments:
>
> > The three relevant papers are Cai et. al. (2019), Xu and Gu (2020) and Cayci et. al. (2021). The authors reference problems in that analysis - that the projection radius needs to shrink at a rate of $O(m^{-1/2})$  -- but don't specifically compare the results in these papers to their results presented in Theorem 1.
>
> We are happy to give a more detailed comparison and are looking forward to the back-and-forth in this conversation. See below a point-by-point response and please also don't miss the ``big picture'' reply at the end.
>
> > My understanding is that most of these papers also characterize the quality of solutions that Neural TD converges to - for example, see section 4 in Cai et. al. (2019) - the approximation error here may be non-vanishing but there is a characterization of the limit point in terms of the solutions of the projected Bellman equation.
>
> Let us look carefully at Section 4 of Cai et al (2019) brought up by the reviewer. That work begins by defining the set of single-hidden-layer neural networks
> $$ \widehat{Q}(x;W) = \frac{1}{\sqrt{m}} \sum_{r=1}^m b_r \sigma \left(W_r^T x \right) $$ but then proceeds by fixing the nonlinearity $\sigma(\cdot)$ to whatever it would evaluate at the initial point:
> $$ \widehat{Q}_0(x;W) =  \frac{1}{\sqrt{m}} \sum_r b_r {\bf 1}(W_r(0)^T x > 0) W_r^T x.$$
> Thus the new function $\widehat{Q}_0(x;W)$ is *linear* in $W$. In particular, $\widehat{Q}_0(x;W) = \phi(x)^T W$, for some nonlinear transformation $\phi(\cdot)$.
>
> Next comes the definition of an approximate stationary point $W^*$, which is defined as the stationary point of the update on the new linear function $\widehat{Q}_0(x;W)$. Then an assumption is made which essentially says that, under the stationary distribution, the state has a positive density over the unit sphere (this is Assumption 4.3; note that the probability that the state is almost orthogonal to **any** vector $w$ has to go zero with the measure of orthogonality, so no part of the unit sphere can be ``missed''). We note that this is a strong assumption which one does not expect to be satisfied in practice, where states may concentrate in a-priori unpredictable ways.  Finally, Theorem 4.4 gives a result that says that the output of neural TD is not too far away from $\widehat{Q}_0(x;W^*)$, i.e., the linear map at its stationary point.
>
> It should be clear from the above discussion that this result is essentially linear. It raises the natural question: *why should we do neural TD in the first place?* Why not simply linearize around the initial point and then do linear TD? After all, in the final result here, these two possibilities give essentially the same answer.
>
> We hope it is clear why this is not a fully satisfying result: in practice, people use neural networks in RL, and they don't just use random linear features! *We would suggest that whatever assumptions lead to the conclusion that neural TD is not far from linear TD are not the most interesting assumptions to consider.* In Cai et al.  (2019) what seems to be responsible for this is the combination of random initialization and the uniform randomness of the state under the stationary distribution -- indeed, tracing that proof down to Lemma H.1 suggests that, under these conditions, the nonlinearity does not have much of an effect on the average over a bounded domain.
>
> Our goal, therefore, is to provide an analysis of neural TD which, at the very least, is not immediately reducible to the linear case. We'll come back to the question of whether we achieve this below.
>
> >

---

> ### Author Response · Authors · 2022-11-14
> **Reply to reviewer oRiv: part II**
>
> > The results in Cayci et. al. (2021) seems stronger. In addition to provide a projection free analysis, they also give guidance about scaling the network width for a given value of target error. It seems that in their results,
> $\epsilon$ can be arbitrary large and there is now guidance on how to select $\omega$.
>
> There are a number of ways in which our result improves upon the previous work of Cayci et al (2021):
>
> * Cayci et al. consider a single hidden layer. We allow any number of layers.
>
> * Cayci et al. make the following representability assumption. They define the function
> $$ \phi(x;w) = {\bf 1}(w^T x \geq 0) x.$$
> They then assume that there exist a function $v(w)$ such that  the true value function $V(x)$ satisfies
> $$V(x) = E_{w \sim N(0,I)} \left[ v(w)^T \phi(x;w) \right].$$ Further, they assume that $\sup_{w} ||v(w)|| \leq \overline{\nu} < \infty$. This is Assumption 2 in their paper, and it goes without saying it is very strong. It is difficult for us to fully grasp what it means -- though see the discussion in their paper about how this assumption is not far from assuming to the true value function $V(x)$ belongs  to a certain RKHS. *By contrast,  we make no assumptions at all on representability of the true value function (note that $\epsilon$, the approximation quality of the best neural network in approximating the true value function in our paper, can be anything).*
>
> * Cayci et al assume that the width of the network is proportional to the state-space dimension $d$. This is clearly limiting in high-dimensional settings. We do not make this assumption.
> * Similarly, the number of iterations need to achieve the error bound in Cayci et al scales with dimension, whereas ours does not.
> * To achieve an error of $\epsilon$, Cayci et al. require a width of $ m \geq O(\epsilon^{-4})$ (note that the parameter $\lambda$ in that paper depends on $\epsilon$, which is where the extra scalings with $\epsilon^{-1}$ come in). By contrast, if we assume that there is a neural-network that represents the true value function, our error of $O(m^{-1/2})$ translates into a much better bound $m \geq O(\epsilon^{-2}) $.
> * The projected analyses in Cayci et all assumes projection onto a ball of radius $\sim m^{-1/2}$ around the initial condition. This effectively keeps the neural network around initialization, which we do not do.
> * **Most important point:** The projection-free method in Cayci et al. assumes a lower bound on width under which the ``lazy training'' regime applies, i.e., neural TD always stays within $\sim 1/\sqrt{m}$ of its initialization (last line of the statement of Theorem 1).
> This allows them to avoid a projection but makes the nonlinearity of the analysis once again questionable (when the network stays close to it's initial condition, it is well-aproximated by the linearization around the initial condition as discussed in Cai et al (2019) above). The most interesting cases of neural TD analysis should allow the network to actually change weights during training to find good features, and not not to constrain it to stay close to where it started.
>
> > Maybe an empirical comparison to projection free and max-norm scaling methods of Cayci et. al. (2021) is also useful.
>
> Note that the max-norm method of Cayci et al. just does projection on a range of $1/\sqrt{m}$ around the initial point, which we already compare to our method -- and the results show that this projection onto a small ball strongly deteriorates performance relative to a constant projection radius.
>
> Here is an additional simulation to buttress this point. We plot the distance from the initial point over training, divided by the projection radius:
>
> https://anonymous.4open.science/r/images_in_paper-87D6/dist.png
>
> In every single case, both for constant and decaying projection radius, we see neural TD moves to a point on the *boundary* of the set we are projecting onto. This further substantiates the claim that decaying projection radii are extremely restrictive.
>
> Direct comparison against the projection-free algorithm in Cayci et al. is difficult, since that algorithm requires $m$ to be larger than the values we simulate. When we simulate it for smaller $m$ like the ones we use in our paper, we often encounter divergence. It appears that a projection is necessary to stabilize the algorithm.

---

> ### Author Response · Authors · 2022-11-14
> **Reply to reviewer oRiv: part III**
>
> > I request the authors to provide a more detailed, clear and transparent comparison to past work - highlighting the strengths and weaknesses of their approach as well as results vis-a-vis past work. That would be immensely useful.
>
> We have given this analysis above for the two papers the reviewer requested (and the comparison to the third paper Xu & Gu (2020) is essentially the same). But now let us focus on the "big picture."
>
> We were motivated to write this paper because we were not with fully satisfied with the state-of-the-art in theory of neural TD: results seem to only work by guaranteeing things stay close to the initialization, in which case the benefit of using neural vs. linear TD is questionable; further, all results seem to have strong representability assumptions that seem unlikely to be satisfied in practice.
>
> What we wanted to obtain was a simple theorem: if there exists a NN that yields an $\epsilon$-approximation of the true value function, then neural TD finds an approximation whose quality is
>
> $\epsilon + $ something that decays with iteration $T$  $+$ something that decays with width $m$,
>
> ...and without any representability conditions, and without any assumptions that force the NN to stay close to the initial condition.**This is exactly what our main result does.** Our simulations further confirm that, on every example we tried, the network does not stay close to the initial condition, in fact moves to the boundary of the projection region, and outperforms networks which are restricted to stay close to where they started. We hope the contribution of our work is clear.
>
> > While the main paper is well written, I think the Appendix requires some work in clear writing. Instead of saying 'By Lemma A.5 and eq(13)' can you kindly write out each expression and bound term by term? Can the equations be properly numbered to show how $g(\theta)$ can be
>  can be rewritten as' .. Similar comments go for the proof of non i.i.d case. I assume most inequalities to be correct (since the resulting bounds make sense), I don't think these have been properly justified. I think it is really bad to put the onus on reviewers to parse badly written proofs - the burden must be on the authors to write in a clear and easy to follow manner.
>
> Thank you for pointing this out.  We have gone through the appendix again and added more details to many points of the proofs. For example, instead of saying 'By Lemma A.5 and eq(13)', we now give more detailed steps of how the bound was obtained.

---

> ### Author Response · Authors · 2022-11-14
> **Reply to reviewer oRiv: part IV**
>
> Looking over our earlier replies, it occurs to us that the reviewer asked us to give a detailed comparison against all the previous literature, whereas we compared to the 2/3 papers Cai et al. and Cayci et al. Thus, we next make a detailed comparison to the last paper, which is Xu & Gu:
>
> *  Xu & Gu use a projection radius that scales as $m^{-1/2}$ around the initial point, which keeps the neural network close to where it started. By contrast, we use a constant projection radius.
>
> * Relatedly, Xu & Gu compare their performance to the performance of a certain linearization around the initial point. The linearization is the same as used in Cai et al and already discussed in part I of our response. By contrast, we compare to the best neural network in the constant-radius ball around the initial condition.
>
> *   Xu & Gu make an assumption about the policy they are evaluating: quoting from their paper, "we essentially require that the learning policy $\pi$ is not too bad compared with the greedy policy." The exact form of that assumption may be found as Assumption 5.3 in their work. They also mention that this assumption can be relaxed to assuming regularity of the state as done by Cai et al (and already discussed by us in part I of the response). By contrast, we make no assumptions about the policy.
>
> * *Minor point:* The bound given in Xu & Go actually diverges as $T \rightarrow \infty$ given the $\log(T)$ in the final term. Therefore, making that bound small requires so-called "early stopping," i.e., choosing $T$ judiciously based on the other parameters. By contrast, our result allows $T$ to be arbitrary, and in particular allows running neural TD for an arbitrarily long time.
>
> * *Minor point:* The width of the neural network in Xu & Gu has to scale with the dimension, whereas ours does not.
>
> * *Minor point:* The error in Xu & Gu scales as $m^{-1/6}$ in the width, whereas our error scales as $m^{-1/2}$.
>
> In general, we reiterate that the major innovation in our result is that it is the first analysis where the network is allowed to significantly deviate from the initial condition, whereas previous works did not allow this either by introducing a projection, or making certain assumptions which ensured the network stayed close to initialization. This is the first and second point in the bulleted list above. Note that the major allure of using neural networks as opposed to linear approximations is precisely so that the network moves away from its initial weights.
>
> Finally, while we understand how our result might seem straightforward at first glance, please note that we are significantly improving on the work of several top-tier research groups which only analyzed neural TD with important caveats. Our core conceptual argument is indeed a simple one ("TD is gradient splitting, and the nonlinearity in neural TD results in an error term in the resulting SGD-type recursion that scales as $m^{-1/2}$, translating into an additive term in the final error") but the fact that it leads to a compelling improvement over the state-of-the-art suggests that, in this case, a conceptually simple idea is not necessarily straightforward to discover and execute.

---

### Decision · Program_Chairs · 2023-01-20

**Decision:**

Accept: poster

**Justification For Why Not Higher Score:**

Reviewers felt that the work is somewhat incremental.


**Justification For Why Not Lower Score:**

The work fills a gap in the literature.

**Metareview: Summary, Strengths And Weaknesses:**

The paper provides a convergence analysis of Neural TD Learning, an approximate temporal difference method for policy evaluation that uses a neural network for function approximation. The authors provide an approximation bound. The reviewers generally appreciated the work and found the proofs technically correct. TD learning is core to reinforcement learning, and its extension to neural networks enhances it to a more practical and powerful setting. Convergence analysis of this algorithm is an important contribution to the community; likewise, I recommend accepting this paper.


**Note From Pc:**

if the above contains the word "oral" or "spotlight" please see: "oral" presentation means -> notable-top-5% and "spotlight" means -> notable-top-25%. As stated in our emails, we are disassociating presentation type from AC recommendations